# Turnip mosaic virus P1 suppresses JA biosynthesis by degrading cpSRP54 that delivers AOCs onto the thylakoid membrane to facilitate viral infection

**Mengfei Ji**[1,2,3], **Jinping Zhao**[3], **Kelei Han**[2], **Weijun Cui**[1], **Xinyang Wu**[1], **Binghua Chen**[2], **Yuwen Lu**[2,3], **Jiejun Peng**[2,3], **Hongying Zheng**[2,3], **Shaofei Rao**[2,3], **Guanwei Wu**[2,3], **Jianping Chen**[1,2,3]*, **Fei Yan**[2,3]*

**1** College of Agriculture and Biotechnology, Zhejiang University, Hangzhou, China, **2** State Key Laboratory for Managing Biotic and Chemical Threats to the Quality and Safety of Agro-products, Institute of Plant Virology, Ningbo University, Ningbo, China, **3** Key Laboratory of Biotechnology in Plant Protection of MOA and Zhejiang Province, Institute of Virology and Biotechnology, Zhejiang Academy of Agricultural Sciences, Hangzhou, China

* jianpingchen@nbu.edu.cn (JC); yanfei@nbu.edu.cn (FY)

## Abstract

Jasmonic acid (JA) is a crucial hormone in plant antiviral immunity. Increasing evidence shows that viruses counter this host immune response by interfering with JA biosynthesis and signaling. However, the mechanism by which viruses affect JA biosynthesis is still largely unexplored. Here, we show that a highly conserved chloroplast protein cpSRP54 was downregulated in *Nicotiana benthamiana* infected by turnip mosaic virus (TuMV). Its silencing facilitated TuMV infection. Furthermore, cpSRP54 interacted with allene oxide cyclases (AOCs), key JA biosynthesis enzymes, and was responsible for delivering AOCs onto the thylakoid membrane (TM). Interestingly, TuMV P1 protein interacted with cpSRP54 and mediated its degradation via the 26S proteosome and autophagy pathways. The results suggest that TuMV has evolved a strategy, through the inhibition of cpSRP54 and its delivery of AOCs to the TM, to suppress JA biosynthesis and enhance viral infection. Interaction between cpSRP54 and AOCs was shown to be conserved in Arabidopsis and rice, while cpSRP54 also interacted with, and was degraded by, pepper mild mottle virus (PMMoV) 126 kDa protein and potato virus X (PVX) p25 protein, indicating that suppression of cpSRP54 may be a common mechanism used by viruses to counter the antiviral JA pathway.

## Author summary

Jasmonic acid pathway has emerged as one of the predominant battlefields between plants and viruses. Several studies have indicated that, in addition to interfering with JA signaling, plant viruses can also affect JA biosynthesis, but the direct molecular links between them remain elusive. Here, we identify a highly conserved chloroplast protein cpSRP54 as

**Data Availability Statement:** All relevant data are within the manuscript and its Supporting Information files.

**Funding:** This work was supported by China Agriculture Research System of MOF and MARA (CARS-24-C-04) awarded to F.Y., the National Natural Science Foundation of China awarded to S. R. (31901849) and to G.W. (32070165), and the K. C. Wong Magna Fund of Ningbo University awarded to F.Y. The funders had no role in study design, data collection and analysis, decision to publish, or preparation of the manuscript.

**Competing interests:** The authors have declared that no competing interests exist.

a key positive regulator in JA biosynthesis and a common target for viruses belong to different genera. Through associating with cpSRP54 and inducing its degradation using the protein they encoded, the viruses can inhibit the cpSRP54-facilitated delivery of AOCs to the thylakoid membrane and manipulation of JA-mediated defense. This capability of viruses might define a novel and effective strategy against the antiviral JA pathway.

## Introduction

The lipid-based hormone jasmonic acid (JA) is an essential signaling molecule during plant development and in plant stress responses [1]. JA biosynthesis is initiated from α-linolenic acid (α-LeA, 18:3) by sequential catalyzation of 13-lipoxygenase (13LOX), allene oxide synthase (AOS) and allene oxide cyclase (AOC), leading to the formation of the JA precursor OPDA within chloroplasts. OPDA is then transported into peroxisomes where it undergoes reduction by OPDA reductase 3 (OPR3) and three subsequent steps of β–oxidization to generate JA [2]. The JA receptor CORONATINE INSENSITIVE 1 (COI1), which forms the SCF$^{COI1}$ complex, degrades JA ZIM (JAZ) proteins through the 26S proteosome, thus releasing the repression of downstream transcription factors such as MYC2, MYB21 and ORA59, and inducing the expression of defense-related genes like *PDF1.2*, *PR3* and *PR4* [1,3].

JA plays a vital role in plant antiviral defense [4,5]. Exogenous application of methyl jasmonate (MeJA), a volatile methyl ester of JA, efficiently induces resistance in *Arabidopsis thaliana* to beet curly top virus (BCTV, genus *Begomovirus*), in *Nicotiana benthamiana* to tobacco mosaic virus (TMV, genus *Tobamovirus*) and tomato spotted wilt virus (TSWV, genus *Tospovirus*) and in rice to rice ragged stunt virus (RRSV, genus *Oryzavirus*), rice stripe virus (RSV, genus *Tenuivirus*) and rice black streaked dwarf virus (RBSDV, genus *Fijivirus*) [6–12]. Conversely, inhibition of JA signaling is beneficial for viral infection: the JA-insensitive mutant *coi1* and *myc3* of rice are more susceptible to rice-infecting viruses including RBSDV and RSV than the wild type [11,13,14].

Increasing evidence shows that viruses counter the plant JA-mediated defense by interfering with the JA pathway, especially JA signaling, to enhance viral infection [5]. For example, cucumber mosaic virus (CMV; genus *Cucumovirus*) 2b protein competitively binds to JAZ proteins with COI1, the receptor of JA, preventing JA-induced degradation of JAZ repressors, thereby inhibiting JA signaling [15]. βC1 of tomato yellow leaf curl China virus (TYLCCNV; genus *Begomovirus*) interacts with MYC2 and interferes with its dimerization, thus inhibiting the JA-mediate response [16]. Several rice-infecting viruses encode transcription repressors to disassociate the MED25-MYC complex and cooperate with JAZ to improve their transcriptional repression activity, thus inhibit JA signaling [14]. Hijacking the ubiquitin proteasome system has also been shown to be a common strategy adopted by viruses to repress JA signaling. Both the C2 protein of tomato yellow leaf curl virus (TYLCV; genus *Begomovirus*) and the RBSDV P5-1 protein inhibit the ubiquitination activity of SCF E3 ligases, thus affecting the transcription of response genes in the JA signaling pathway [10,17].

Several reports have shown that the expression of genes involved in JA biosynthesis is suppressed in plants infected by RRSV, RBSDV and tomato yellow leaf curl Sardinia virus (TYLCSV, genus *Begomovirus*), which suggests that JA biosynthesis can also be targeted by viruses to enhance viral infection [7,12]. However, it is not yet clear how this regulation operates. We now show that the P1 protein of turnip mosaic virus (TuMV, genus *Potyvirus*) mediates the degradation of cpSRP54 through the 26S proteosome and autophagy pathways. cpSRP54 is described as an essential component of a chloroplast translocation system cpSRP,

which is required for efficient targeting of many thylakoid membrane proteins [18]. Here, we found that cpSRP54 is responsible for localizing AOCs, key enzymes for JA biosynthesis, onto the thylakoid membrane (TM), suggesting a mechanism by which TuMV suppresses JA biosynthesis to enhance its infection. We also demonstrate that cpSRP54 interacts with AOC in Arabidopsis and rice, and that cpSRP54 can also be repressed by pepper mild mottle virus (PMMoV; genus *Tobamovirus*) and potato virus X (PVX; genus *Potexvirus*) in *N. benthamiana*, indicating that cpSRP54 may be a common target that viruses manipulate to inhibit JA-mediated defense.

## Results

### cpSRP54 is downregulated in TuMV-infected *Nicotiana benthamiana* and its silencing facilitates viral infection

cpSRP54 is a highly conserved 54-kDa chloroplast signal recognition particle subunit which is required for many key photosynthetic proteins to target thylakoid membranes (TM) [18]. The cpSRP54 of *N. benthamiana* has respectively 71.7%, 67.7% and 80.4% amino acid identity with two rice cpSRP54 homologues, cpSRP54a (OscpSRP54a, accession no. BAT12736.1), cpSRP54b (OscpSRP54b, accession no. ABG22369), and Arabidopsis cpSRP54 (AtcpSRP54, accession no. AAC64139.1) (S1A and S1B Fig). Our previous label-free quantitative proteomics analysis had shown that the accumulation of cpSRP54 was significantly reduced by RSV infection in *N. benthamiana* [19] and in this study cpSRP54 protein accumulated at a lower level in *N. benthamiana* leaves infected by TuMV than in the mock-inoculated controls, suggesting the downregulation of cpSRP54 in TuMV-infected *N. benthamiana* (Fig 1A). Consistently, *cpSRP54* transcripts were also down-regulated in TuMV-infected plants (S1C Fig).

To determine the potential biological function of cpSRP54 during TuMV infection, we silenced *cpSRP54* in *N. benthamiana* plants using the tobacco rattle virus (TRV)-induced gene silencing system (VIGS) and then inoculated plants with a modified clone of TuMV expressing the green fluorescent protein (TuMV-GFP) to monitor infection. To silence *cpSRP54*, a partial sequence of *cpSRP54* was inserted into TRV RNA2, producing TRV:cpSRP54, according to the method reported previously [20]. The empty TRV vector (TRV:00) was used as a control. At 12 dpi, TRV:cpSRP54 treatment caused chlorosis in leaves (Fig 1B) and the levels of *cpSRP54* transcripts in them were only 5% of those in the TRV:00-treated controls (Fig 1C). Correspondingly, the protein levels of cpSRP54 were significantly less in the TRV:cpSRP54-treated plants than in the non-silenced control plants (Fig 1D). These results confirm the effective silencing of *cpSRP54*. TuMV-GFP was then inoculated onto the plants. At 6 dpi of TuMV-GFP, green fluorescence appeared under UV light in the top leaves of both *cpSRP54*-silenced and non-silenced plants, indicating systemic infection by TuMV-GFP, but the fluorescence was more intensive and extensive on the silenced plants (Fig 1E). Consistently, TuMV CP accumulated at a higher level in *cpSRP54*-silenced leaves (Fig 1F and 1G).

To minimize any effect of the chlorosis caused by TRV:cpSRP54-inoculation on TuMV infection, we silenced *cpSRP54* by transiently expressing the *cpSRP54* hairpin RNAi construct. Leaves agroinfiltrated with *cpSRP54* hairpin RNAi construct alone had reduced mRNA levels of *cpSRP54* (94%) and did not show obvious chlorosis at 3 dpi (S2A and S2B Fig). Silenced leaves were then mechanically inoculated with TuMV-GFP. At 7 dpi of TuMV-GFP, TuMV CP accumulated to a higher level in the top leaves of plants treated with *cpSRP54* hairpin RNAi construct compared with those treated with the control hairpin construct targeting the unrelated *β-glucuronidase* (*GUS*) gene, which indicates that the effect of *cpSRP54* silencing on TuMV infection was not due to chlorosis (S2C and S2D Fig).

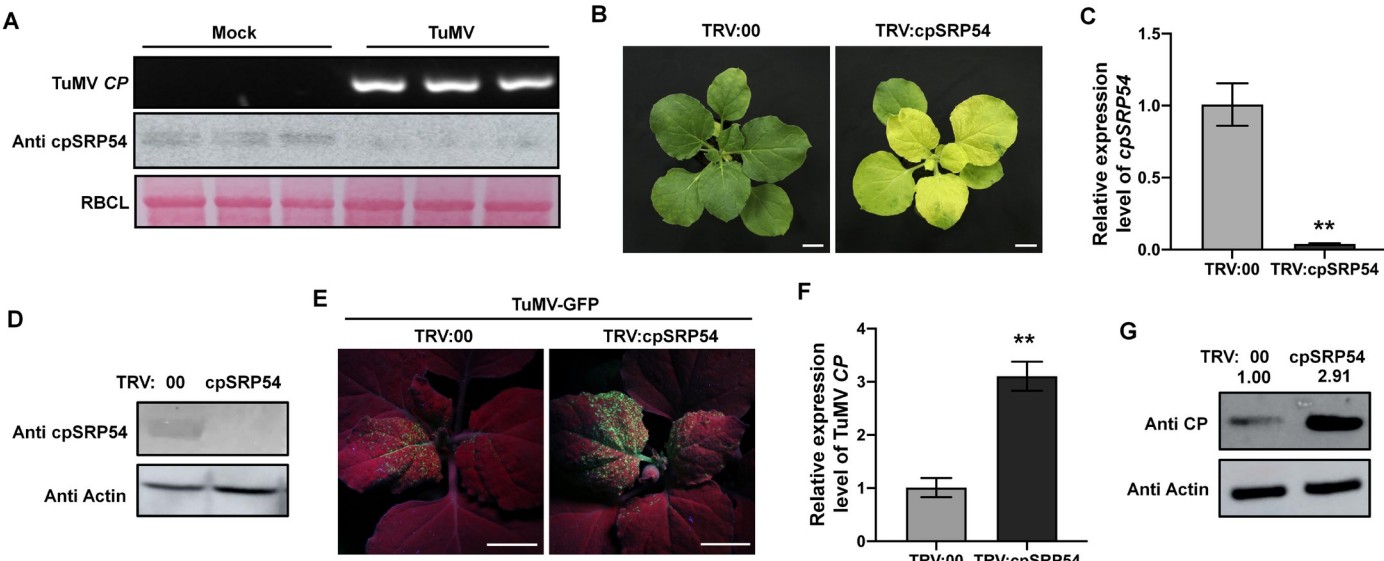

**Fig 1. Silencing of *cpSRP54* facilitates TuMV infection.** A. TuMV *coat protein* (*CP*) gene was detected by RT-PCR to confirm the infection of TuMV in plants at 6 dpi. The accumulation of cpSRP54 proteins in TuMV-infected plants was detected by its antibody through western blot (WB) analysis. Ponceau S-stained RBCL was used as a loading control. B. Phenotype of TRV:00 or TRV:cpSRP54 treated plants at 12 dpi. TRV:cpSRP54 treated plants showed chlorosis on the top leaves but no other developmental defects. Scale bar, 2 cm. C. Quantification of *cpSRP54* mRNA levels in TRV: cpSRP54 treated plants at 12 dpi by qRT-PCR analysis. Means ± SD values are from three independent plants per treatment and were normalized against *Actin*. **, P<0.01 according to Student's *t*-test. D. cpSRP54 protein level in TRV:cpSRP54 treated plants was detected by WB analysis using its antibody at 12 dpi. Actin served as a loading control. Tests were independently performed three times with similar results. E. GFP fluorescence on the newly-emerged leaves of plants pretreated with TRV:00 or TRV:cpSRP54 then infected by TuMV-GFP. Plants were photographed under UV light at 6 dpi. Scale bar, 2 cm. F. Viral RNA was assessed by qRT-PCR, using *Actin* as an internal control. Means ± SD values are from three independent plants per treatment. **, P<0.01 according to Student's *t*-test. G. Viral CP accumulation was detected by WB assay using CP antibodies.

Taken together, the results demonstrate that TuMV infection downregulates cpSRP54 expression and that silencing of *cpSRP54* facilitates viral infection.

## JA is reduced in *cpSRP54*-silenced plants and MeJA treatment alleviates the susceptibility of *cpSRP54*-silenced *N. benthamiana* to TuMV

Chloroplast-derived hormones, including JA, salicylic acid (SA) and abscisic acid (ABA), play a vital role in virus-plant interactions [4,21–23]. We therefore investigated whether these hormones were affected in *cpSRP54*-silenced plants. The JA content in *cpSRP54*-silenced plants was significantly lower than that in non-silenced plants (Fig 2A) whereas SA and ABA were not greatly affected by the reduced expression of cpSRP54 (S3A and S3B Fig). Consistently, transcription levels of the JA-responsive genes, *PDF1.2*, *PR3* and *PR4* [24,25], were downregulated in *cpSRP54*-silenced plants (Fig 2B). These results demonstrate the suppression of the JA pathway in *cpSRP54*-silenced plants.

We next examined whether JA plays a role during TuMV infection. First, we treated plants with 50 μM MeJA for 24 hours and then inoculated them with TuMV-GFP. At 7 dpi, MeJA-treated plants had less TuMV CP than the control plants pretreated with 0.1% ethanol, indicating that JA plays a defensive role against TuMV in *N. benthamiana* (Fig 2C and 2D). To further confirm this, we silenced the *AOCs* which encode allene oxide cyclases that catalyze the cyclization of highly unstable 12, 13-epoxy-octadecatrienoic acid to the JA precursor OPDA during JA biosynthesis [26,27], and monitored TuMV-GFP infection on the silenced plants. In the *N. benthamiana* genome, there are two homologs of AOC with 91.5% amino acid identity, named AOC.1 (Sequence ID in Sol Genomics Network: Niben101Scf02772g02001.1) and

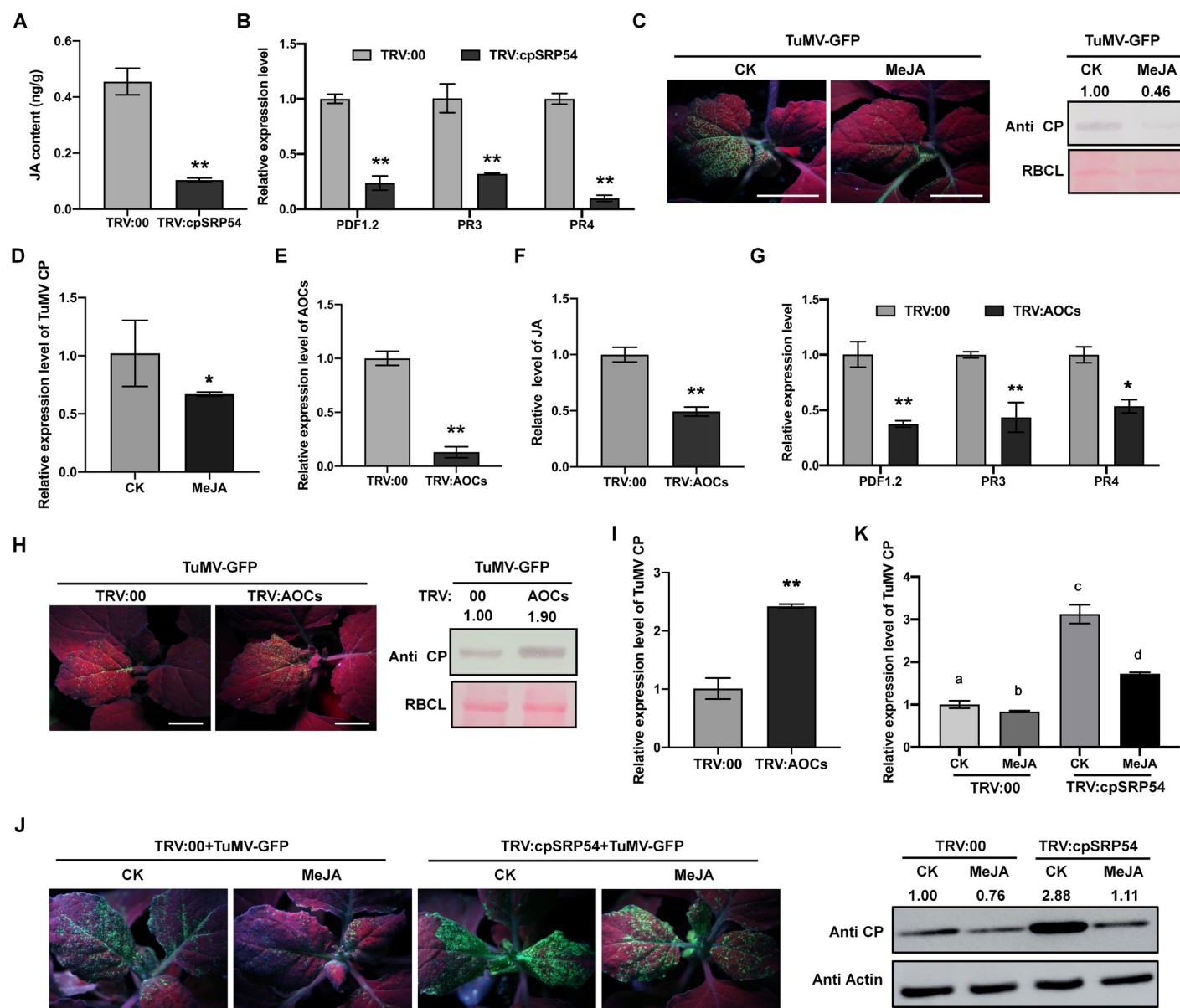

**Fig 2. cpSRP54 is associated with jasmonic acid (JA) that regulates TuMV infection negatively.** A. JA content quantified in TRV:00 or TRV:cpSRP54 treated plants at 12 dpi. **, P<0.01 according to Student's *t*-test. B. Relative expression level of *PDF1.2*, *PR3*, and *PR4* by qRT-PCR. Data are the means ± SD of three biological replicates from each treatment. *Actin* was used as the normalizer. **, P<0.01 according to Student's *t*-test. C. TuMV-GFP infection of plants treated with MeJA or CK (0.1% ethanol) at 6 dpi. Plants were photographed under UV light. Viral CP accumulation was detected by WB. Ponceau S-stained RBCL was used as a loading control. Tests were performed independently three times with similar results. D. Quantification of viral RNA levels by qRT-PCR. Values are shown as means ± SD relative to CK treated plants. *Actin* was used as the normalizer. **, P<0.01 according to Student's *t*-test. E. qRT-PCR analysis confirming the silencing of *AOCs* at 12 dpi. F and G. Relative levels of JA content (F) and JA-responsive genes (G) in plants inoculated with TRV:00 or TRV:AOCs at 12 dpi. **, P<0.01, *, P<0.05, according to Student's *t*-test. H. GFP fluorescence and viral accumulation in plants inoculated with TRV:00 or TRV:AOCs at 6 dpi of TuMV-GFP. Viral CP accumulation was detected by WB. Ponceau S-stained RBCL was used as a loading control. Tests were performed independently three times with similar results. I. Quantification of viral RNA levels by qRT-PCR. Values are shown as means ± SD relative to TRV:00 treated plants. J. Effect of MeJA treatment on TuMV-infection in plants inoculated with TRV:00 or TRV:cpSRP54 at 7 dpi. Viral CP accumulation was determined by WB. Actin served as a loading control. Tests were performed independently three times with similar results. K. Relative level of viral RNA by qRT-PCR analysis. Different letter on histograms indicated significant differences (P<0.05). The protein levels were quantified by ImageJ. Bars, 2 cm.

AOC.2 (Niben101Scf13816g00005.1). The sequences were both silenced simultaneously by VIGS as a result of their high sequence identity (Fig 2E). Plants in which *AOCs* were silenced, had less JA accumulation (~50%) and decreased expression of the JA-responsive genes without

inducing obvious phenotypes (Figs 2F, 2G and S4A) and supported higher accumulation of TuMV (Fig 2H and 2I). Exogenous application of MeJA alleviated the susceptibility of *AOCs*-silenced plants to TuMV (S4B–S4D Fig), further indicating the essential role of the JA pathway in defense against TuMV infection.

As expected, MeJA treatment also alleviated the susceptibility of *cpSRP54*-silenced plants to TuMV (Fig 2J and 2K). Thus silencing of *cpSRP54* facilitates TuMV infection by suppressing the defense role of the JA pathway, indicating that cpSRP54 positively regulates the JA pathway.

## AOCs are substrates of cpSRP54 for localization to the thylakoid membrane

Next, we wanted to know how cpSRP54 regulates the JA pathway. cpSRP54 is known to form a high affinity complex with cpSRP43 to transport the nuclear-encoded light-harvesting chlorophyll a/b binding proteins (LHCPs) to the thylakoid membrane (TM) where they normally function [28–30]. JA biosynthesis enzymes such as 13-LOX, AOS, and AOC are largely found associated with the TM [31,32] and we wondered whether these proteins are substrates of cpSRP54. In a yeast two hybrid (Y2H) assay, the AOC.1 and AOC.2 proteins of *N. benthamiana* both interacted with cpSRP54 (Fig 3A) and this interaction was further confirmed by co-immunoprecipitation (Co-IP) and bimolecular-fluorescence complementation (BiFC) assays (Figs 3B, 3C and S5A and S5B). In addition, the co-localization pattern of cpSRP54-mCherry and AOC.1-GFP was similar to the bright spot-like interaction complex seen within chloroplasts in the BiFC assay (Fig 3D).

We next examined the effect of cpSRP54 on localization of AOCs. As expected, AOC.1-GFP co-localized with mCherry-fused *Arabidopsis* OE23 protein (AtOE23-mCherry), a thylakoid marker which is transported by chloroplast twin-arginine translocation (cpTat) pathway [30], and formed fluorescent spots in the chloroplast, indicating the thylakoid localization of AOC.1 (Fig 4A). However, in *cpSRP54*-silenced cells, fewer green punctate structures of AOC.1-GFP were detected (Fig 4A). Immunolocalization under electron microscopy was then used to confirm the results. AOC.1-GFP was expressed transiently in leaves of TRV:00 or TRV:cpSRP54 inoculated plants by agroinfiltration. At 2 dpi of AOC.1-GFP, ultrathin sections of *N. benthamiana* leaves were incubated with anti-GFP antibody and visualized with a gold-conjugated antibody against mouse IgG. The results showed that AOC.1-GFP was abundant in the TM of TRV:00-inoculated plants, but much less so in TRV:cpSRP54-inoculated plants (Fig 4B, 4C and S2 Table). As further confirmation, we isolated thylakoids to detect the accumulation of AOC.1-GFP in cells. The total AOC.1-GFP in protoplasts isolated from *cpSRP54*-silenced cells and non-silenced cells was similar but there was much less AOC.1-GFP in the thylakoid fraction of *cpSRP54*-silenced cells than in non-silenced cells (Fig 4D).

To determine whether the interaction between cpSRP54 and AOCs influences the stability of AOCs, we expressed AOC.1-GFP with cpSRP54-Myc or the control GUS-Myc. Accumulation of AOC.1-GFP was not affected by expression of cpSRP54, which indicates that the interaction between cpSRP54 and AOCs did not influence the stability of AOCs (S6 Fig).

Using a BiFC assay, we also detected an interaction between cpSRP54 and AOC in rice and Arabidopsis, suggesting that cpSRP54-dependent delivery of AOCs is conserved in plants (S7 and S8 Figs). The results thus demonstrate that cpSRP54 is responsible for transporting AOC.1 to TM and indicate that AOC.1 is a newly-identified substrate protein of cpSRP54 for localization at the TM.

## AOCs delivery to the TM is impaired in TuMV-infected cells

Since TuMV infection leads to the downregulation of cpSRP54, we next examined the effect of the virus on cpSRP54-dependant AOCs delivery by confocal microscopy. In a BiFC assay, the

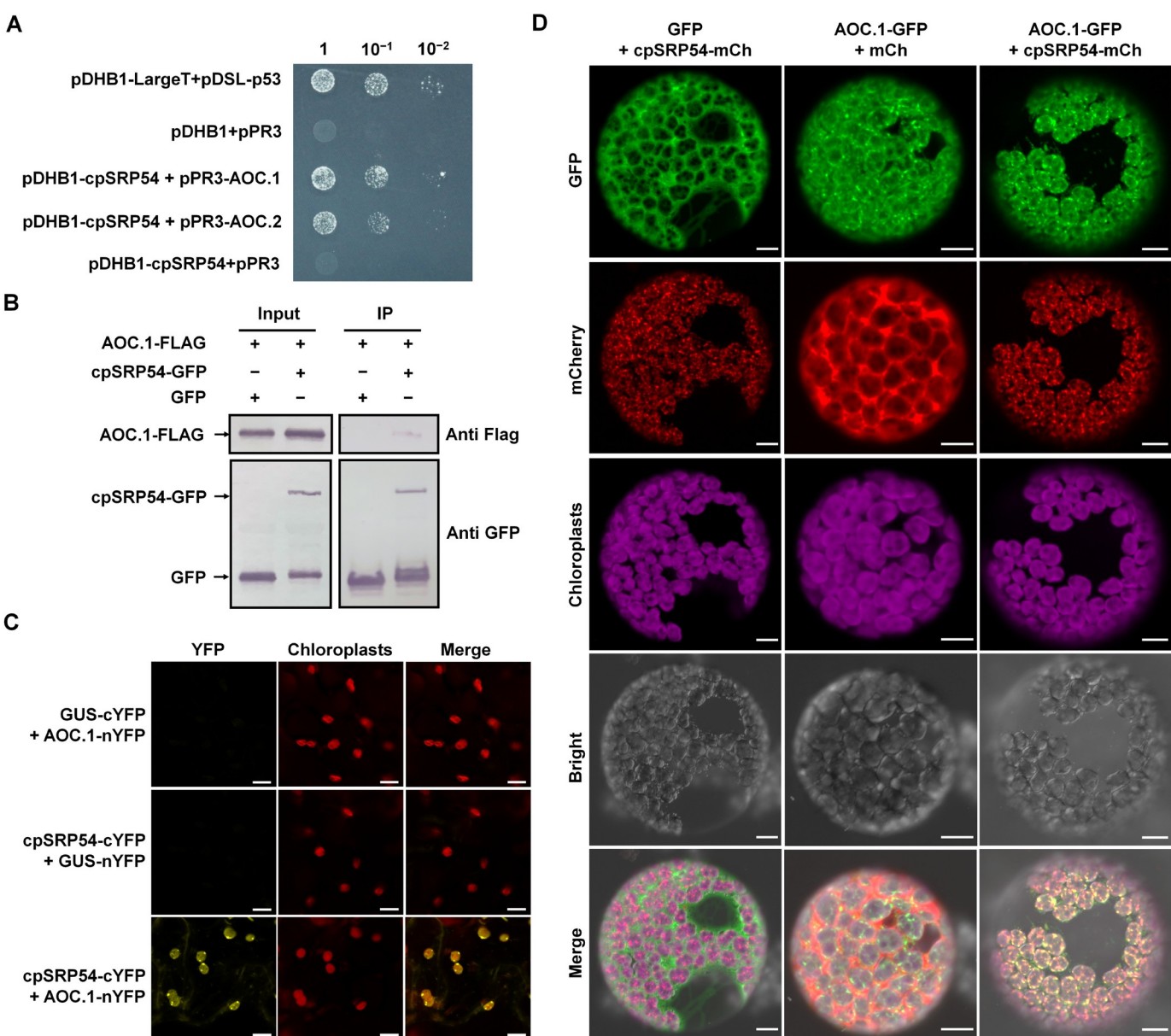

**Fig 3. cpSRP54 interacts with AOC.1.** A. Yeast two hybrid (Y2H) assay showing the interaction between cpSRP54 and AOCs. pDHB1-cpSRP54 and pPR3-AOC.1, pPR3-AOC.2 were co-transformed into NMY51, subjected to 10-fold serial dilutions and plated on synthetic defined (SD) medium (-Ade/-His/-Leu/-Trp). Yeast co-transformed with pDHB1-LargeT and pDSL-p53 served as a positive control and the combinations pDHB1 with pPR3, and pDHB1-cpSRP54 with pPR3 are negative controls. B. Co-immunoprecipitation (Co-IP) assay confirming the interaction between cpSRP54 and AOC.1 in vivo. *N.benthamiana* leaves were co-infiltrated with *Agrobacterium* cultures harbouring expression vectors to express AOC.1-FLAG and GFP (Lane 1), and AOC.1-FLAG and cpSRP54-GFP (Lane 2). Total protein extracts were incubated with GFP beads. Samples before (Input) and after (IP) immunopurification were verified using FLAG and GFP antibody. C. Bimolecular fluorescence complementation (BiFC) confirming the interaction between cpSRP54 and AOC.1. cpSRP54 and AOC.1 were respectively fused to the C-terminal (cYFP) and N-terminal (nYFP) half of YFP. Confocal imaging was performed at 2 dpi. Scale bar, 10 μm. D. Co-localization of cpSRP54-mCherry and AOC.1-GFP in protoplast cells by confocal microscopy at 2 dpi. cpSRP54-mCherry and AOC.1-GFP co-localized with purple chloroplast autofluorescence. Scale bar, 10 μm.

interaction between cpSRP54 and AOCs was not disrupted by viral infection (S9 Fig). When AOC.1-GFP was transiently expressed in TuMV-inoculated *N. benthamiana* there were fewer green punctate structures of AOC.1-GFP, similar to the effect observed in TRV:cpSRP54-inoculated plants. Consistent with this, the accumulation of AOC.1-GFP in TM was reduced (Fig 5A

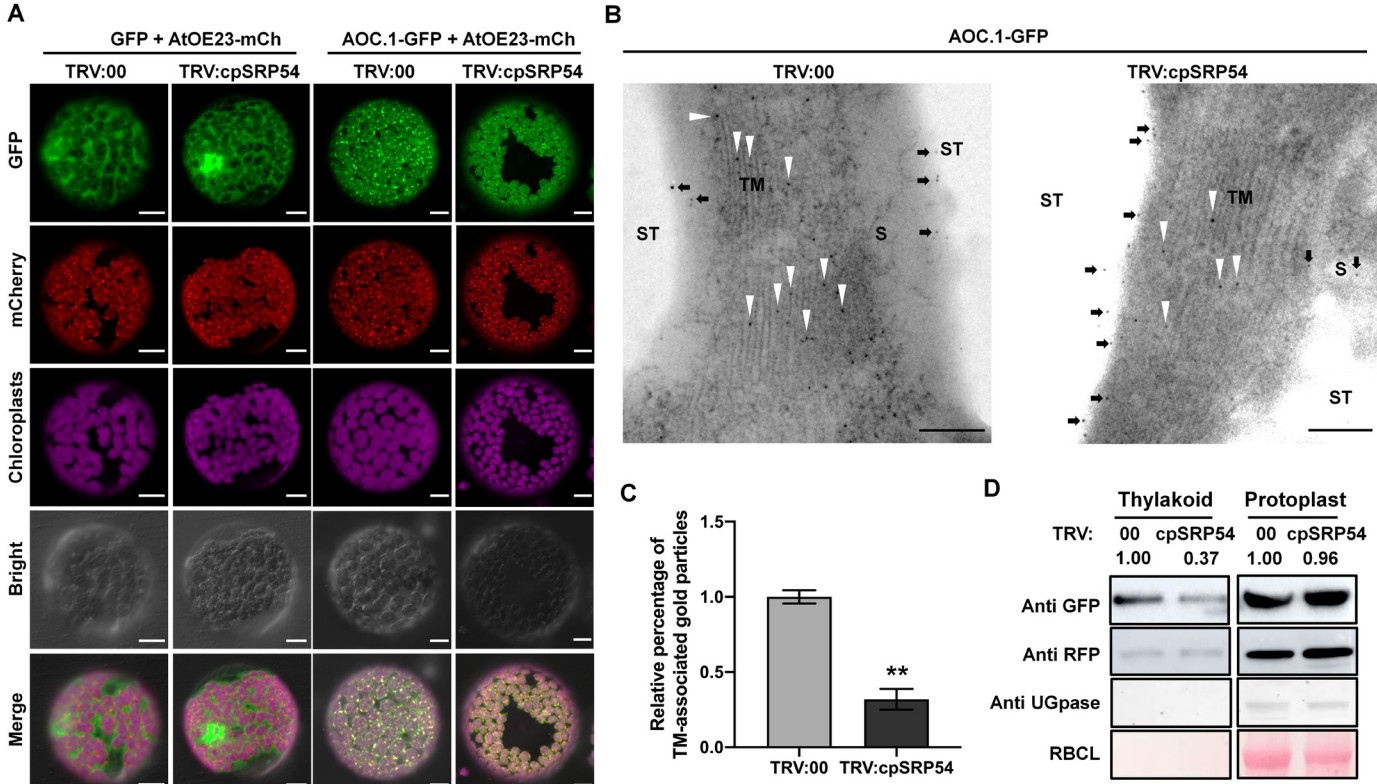

**Fig 4. Silencing of *cpSRP54* changes the localization pattern of AOC.1 within chloroplasts.** A. Co-expression of AOC.1-GFP or free GFP together with mCherry-fused AtOE23 (a thylakoid marker) in *cpSRP54*-silenced or non-silenced *N. benthamiana* protoplasts at 48 hpi. Scale bar, 10 μm. B. Subcellular localization of AOC.1-GFP using immunocytochemistry and electron microscopy. Ultrathin sections were incubated with an anti-GFP antibody and visualized with gold particle anti-mouse antibodies. Arrowheads show some of the gold particles associated with the thylakoid membrane (TM), arrows indicate labelling in the stroma (S). Starch (ST) areas are also shown. Bars, 0.2 μm. C. Percentage of TM-associated AOC.1-GFP labelling in chloroplasts. Gold particles were counted from 4 random chloroplast profiles of the above two treatments. Values represent the average percentage of 4 replicates from each treatment. Error bars represent SD. **, P<0.01 according to Student's *t*-test. D. AOC.1-GFP in protoplasts and thylakoid fractions isolated from *cpSRP54*-silenced or non-silenced *N. benthamiana* plants were detected with GFP antibody by WB analysis. UGPase, UDP-glucose pyrophosphorylase (cytoplasm marker); RBCL (chloroplast stroma marker). The protein levels were quantified by ImageJ and normalized against AtOE23-mCherry protein levels.

and 5B). GFP alone was not affected by TuMV infection (Fig 5A). When AOC.1-GFP was co-expressed with cpSRP54-Myc in TuMV-infected plants, the normal distribution of AOC.1 was partially restored (Figs 5B, 5C and S10). There was also up-regulated expression of transcripts of *AOCs* in TuMV-infected plants (S11 Fig). Taken together, these results indicate that TuMV infection impairs the delivery of AOCs to the TM, possibly by suppressing cpSRP54.

In the period from 3.5 to 14 dpi of TuMV-GFP, the production of JA first increased during the early stages of viral infection, but as viral infection progressed, the relative JA levels in these infected plants declined continuously to reach their lowest level at 9.5 dpi, indicating the ability of TuMV to suppress JA biosynthesis (Fig 5D). Levels increased again at the later stages of TuMV infection presumably because the plants responded to maintain a JA balance (Fig 5D). Therefore, it seems likely from these results that TuMV infection downregulates JA bio-synthesis because it suppresses cpSRP54.

## TuMV P1 interacts with cpSRP54 and degrades it through 26S proteosome and autophagy pathways

Protein-protein interactions between viral proteins and host proteins are one of the main mechanisms used by viruses to establish efficient infection. Although cpSRP54 expression was

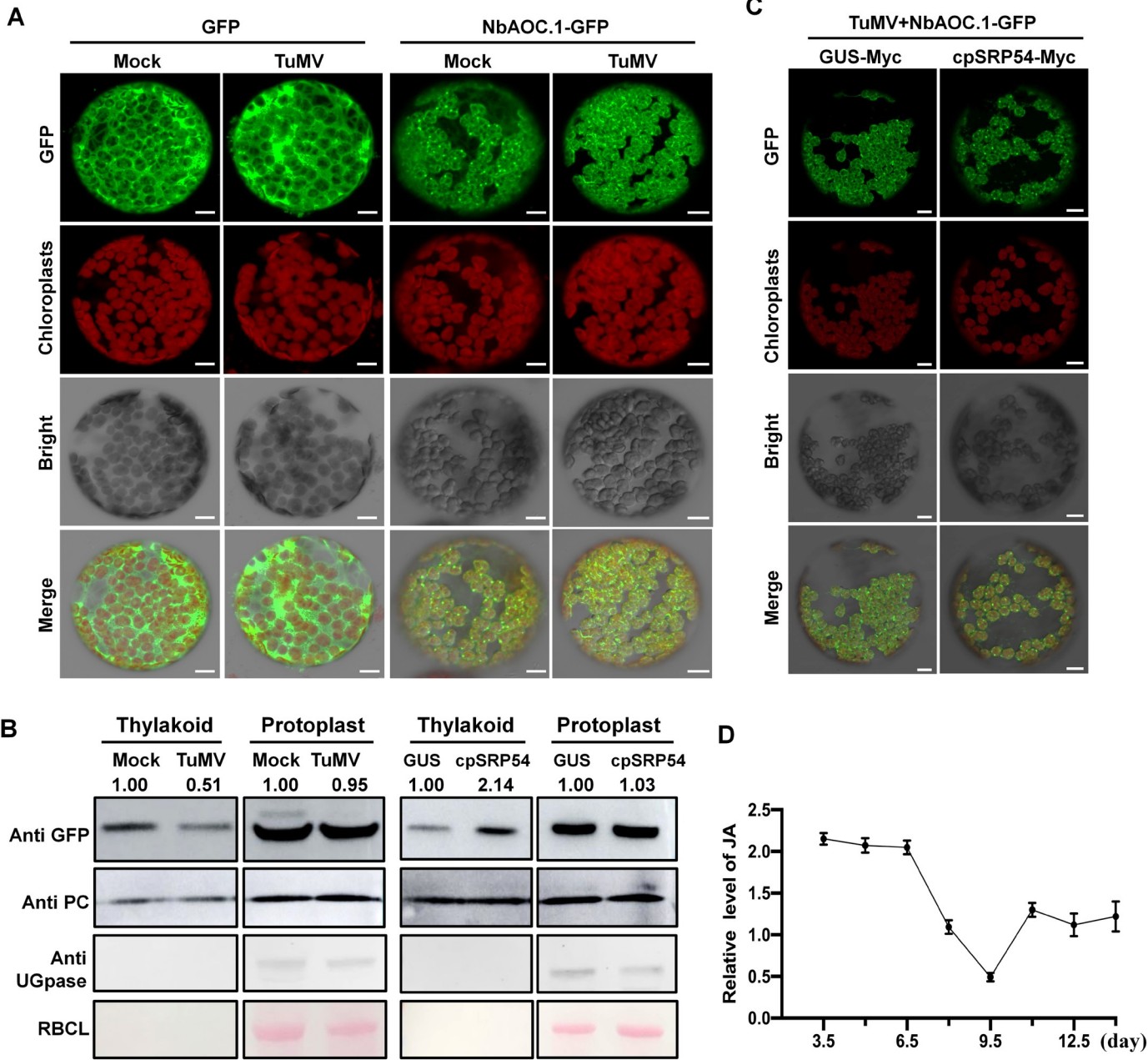

**Fig 5. TuMV infection interferes with the localization of AOC.1.** A. Subcellular localization of AOC.1-GFP and free GFP in protoplasts of mock (inoculated with buffer) and TuMV-infected *N. benthamiana* plants by confocal microscopy at 2 dpi. Scale bar, 10 μm. B. WB of protoplast- and thylakoid-localized AOC.1-GFP transiently expressed in mock and TuMV-infected plants, and in TuMV-infected plants when co-expressed with GUS-Myc or cpSRP54-Myc. Tests were performed twice. C. Localization of AOC.1-GFP in protoplasts of TuMV-infected *N. benthamiana* plants when co-expressed with GUS-Myc or cpSRP54-Myc by confocal microscopy at 2 dpi. Scale bar, 10 μm. D. Relative JA levels in TuMV-infected *N. benthamiana* plants within 14 dpi measured by LC-MS compared to mock plants (agroinfiltrated with empty plasmid). Data are the means ± SD of three biological replicates from each treatment. PC, plastocyanin (thylakoid marker); UGPase, UDP-glucose pyrophosphorylase (cytoplasm marker); RBCL (chloroplast stroma marker). The protein levels were quantified by ImageJ and normalized against PC protein levels.

downregulated by TuMV, we wondered whether TuMV could also directly manipulate cpSRP54. In a BiFC assay, TuMV P1 protein interacted with cpSRP54, and this was confirmed by Y2H and Co-IP assay (Fig 6A–6C). Interestingly, in the Co-IP assay, we noticed that the

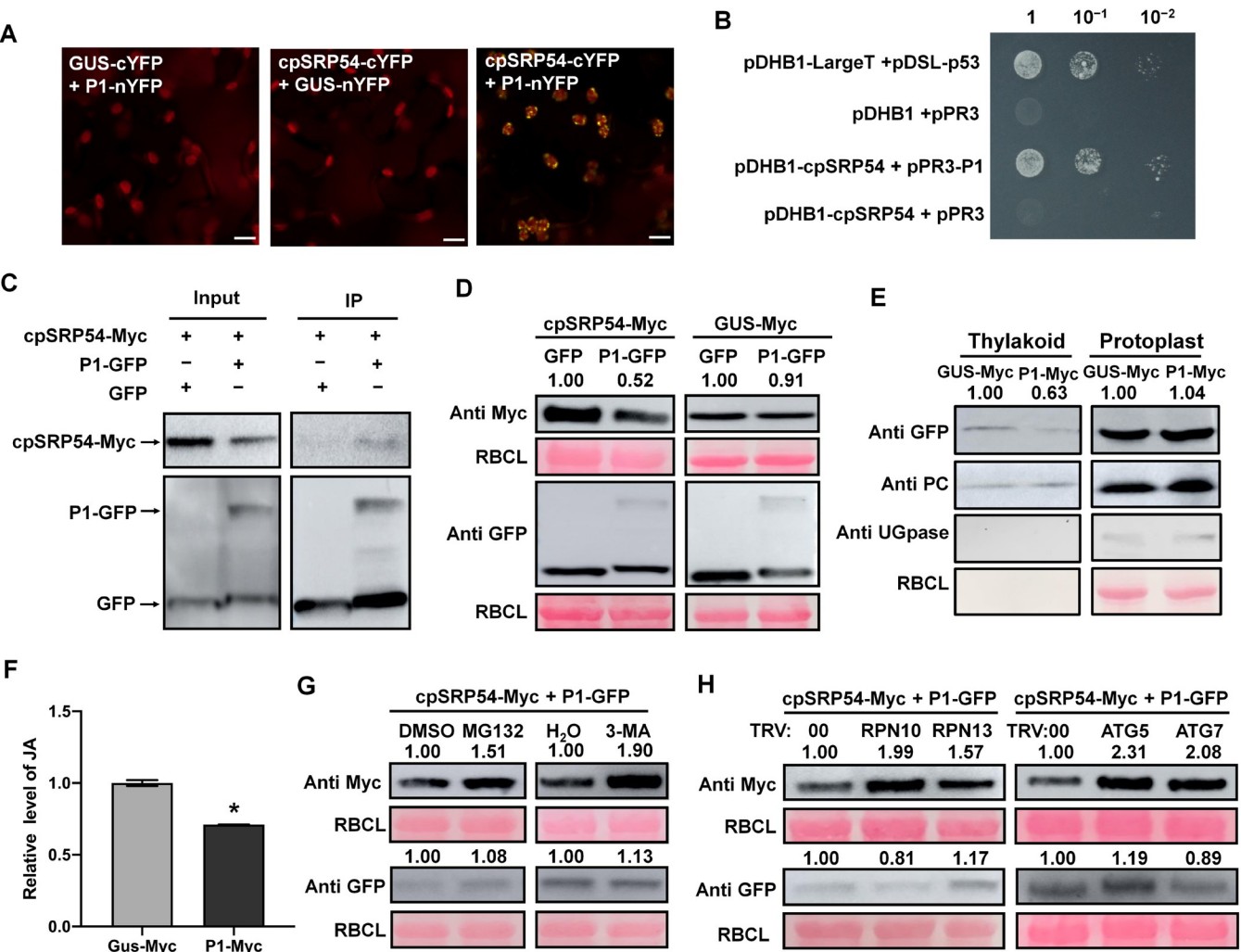

**Fig 6. TuMV P1 protein interacts with cpSRP54 and mediates its degradation through both the 26S proteosome and autophagy pathways.** A. BiFC assay showing the interaction between TuMV P1 protein and cpSRP54 in *N. benthamiana*. Confocal imaging was performed at 2 dpi. Scale bar, 25 μm. B. Y2H assay showing the interaction between P1 and cpSRP54. pDHB1-cpSRP54 and pPR3-P1 were co-transformed into NMY51. pDHB1-LargeT co-transformed with pDSL-p53 was used as a positive control. pDHB1 or pDHB1-cpSRP54 co-transformed with pPR3 into yeast was used as the negative control. A 10-fold series dilutions (1, $10^{-1}$ and $10^{-2}$) are shown from left to right. C. The interaction between cpSRP54-Myc and P1-GFP was confirmed by Co-IP assay. Co-expression of cpSRP54-Myc and GFP serves as a negative control. D. Accumulation of cpSRP54-Myc was reduced by co-expressing P1-GFP. GUS-Myc and empty GFP were used as controls. E. WB of protoplast- and thylakoid-localized AOC.1-GFP transiently expressed in plants when co-expressed with GUS-Myc or P1-Myc at 3 dpi. PC, plastocyanin (thylakoid marker); UGPase, UDP-glucose pyrophosphorylase (cytoplasm marker); RBCL (chloroplast stroma marker). The protein levels were quantified by ImageJ and normalized against PC protein levels. Tests were performed twice. F. Relative JA level in leaves expressing P1-Myc or GUS-Myc at 60 hpi. The endogenous JA in leaves expressing GUS-Myc was set as the baseline. Data are the means ± SD of three biological replicates from each treatment. *, P<0.05. G. Reduction of cpSRP54-Myc caused by expression of P1-GFP was rescued by 26S proteosome inhibitor MG132 and autophagy inhibitor 3-MA. H. Reduction of cpSRP54-Myc caused by expression of P1-GFP was rescued by silencing *RPN10* or *RPN13* (two genes encoding key components of the 26S proteosome), or by silencing *ATG5* or *ATG7* (two genes encoding key components in autophagy). The protein levels were quantified by ImageJ and normalized against RBCL protein levels. Tests were independently repeated three times and representative results are shown.

accumulation of cpSRP54-Myc was obviously reduced when co-expressed with P1-GFP protein (Fig 6C), which suggested that P1 might interfere with cpSRP54 accumulation. To confirm this, cpSRP54-Myc (or GUS-Myc for the controls) was co-expressed with GFP or P1-GFP in leaves of *N. benthamiana*. The protein level of cpSRP54 was obviously decreased when co-expressed with P1-GFP, whereas there were no significant changes in its mRNA level (Figs 6D and S12). In the control experiments, accumulation of GUS-Myc was not affected by P1-GFP,

suggesting that P1 specifically reduced cpSRP54 accumulation (Fig 6D). Potential P1-mediated inhibition of cpSRP54 is also illustrated by the fact that the presence of P1-Myc impaired the cpSRP54-depedant delivery of AOC.1 as well as JA biosynthesis (Figs 6E, 6F and S13). AOC.2 delivery was also blocked in the context of P1 expression as well as TuMV infection (S14 Fig).

There are two major pathways in cells by which proteins are degraded, the 26S proteosome system and autophagy pathway [33–35]. To investigate which pathway was responsible for the reduction of cpSRP54 mediated by P1, further infiltrations were conducted in the presence of either MG132 (an inhibitor of the 26S proteosome system) or 3-methyladenine (3-MA) (an inhibitor of autophagy). The accumulation of cpSRP54-Myc was increased by the co-expression of either MG132 or 3-MA with P1-GFP, while there were no effects of these inhibitors when co-expressing cpSRP54-Myc and GFP, cpSRP54-Myc and CP-GFP, or GUS-Myc and P1-GFP (Figs 6G and S15). The biological relevance of the potential involvement of the two pathways in P1-mediated cpSRP54 degradation was further illustrated by the fact that silencing of genes related to either the 26S proteosome pathway (*RPN10*, *RPN13*) or autophagy (*ATG5*, *ATG7*) increased the accumulation of cpSRP54 when co-expressed with P1-GFP (Figs 6H and S16). Moreover, the thylakoid distribution of AOC.1 and JA level were restored when P1 was expressed in *RPN10*-silenced plants (S17A and S17B Fig). The results suggest that TuMV P1 interacts with cpSRP54 and mediates its degradation through both the 26S proteosome and autophagy pathways, which leads to impaired AOCs delivery and hence, the reduced JA level.

## cpSRP54 is a common target of viruses to inhibit JA-mediated defense

Because the viral RNA silencing suppressors (VSRs) p25 of PVX and 126 kDa protein (p126) of the tobamovirus TMV have previously been shown to inhibit JA-induced gene expression in *N. benthamiana* [36], we investigated whether VSRs like these might act like TuMV P1 and repress cpSRP54 accumulation. To test this possibility, we selected p25 from PVX and p126 from another tobamovirus, PMMoV (two viruses available in our lab), and transiently expressed their RFP fusion proteins, p25-RFP or p126-RFP, to test their effect on the expression of cpSRP54 and JA response genes. According to the qRT-PCR analysis, both VSRs caused a decrease in the mRNA levels of JA response genes *PR3* and *PR4* at 2 dpi, but had no noticeable effect on the mRNA level of *cpSRP54* (S18A and S18B Fig). Interestingly, the western blot assay showed that the protein level of cpSRP54 was markedly decreased with increasing amounts of p25 or p126, indicating that these two unrelated VSRs are capable of triggering cpSRP54 degradation (Fig 7A). A further BiFC assay indicated a potential interaction between cpSRP54 and the two VSRs (Fig 7B).

Consistently, the cpSRP54 protein level was reduced by both PMMoV and PVX as shown by the western blot assay (S19A Fig). The biological relevance of the virus-mediated inhibition of cpSRP54 was also illustrated by the fact that *N. benthamiana* plants in which *cpSRP54* has been knocked down by VIGS were more susceptible to both PMMoV and PVX (S19B and S19C Fig). The defense role of JA during viral infection was also demonstrated by an assay showing that MeJA treatment conferred plant resistance to viruses (S19B and S19C Fig).

## Discussion

JA plays an important role in plant immunity against various pathogens including plant viruses [6,37,38]. JA signaling can induce pathogenesis-related (PR) defense genes such as *PR3*, *PR4* and *PDF1.2*. Upon viral infection, JA has been shown to network with other antiviral defense pathways such as the RNA silencing machinery [13], and plant hormones including brassinosteroid (BR) [11] and SA [8], to mediate defense responses, making plants respond more efficiently to viruses. Viruses in turn have evolved strategies for survival that include

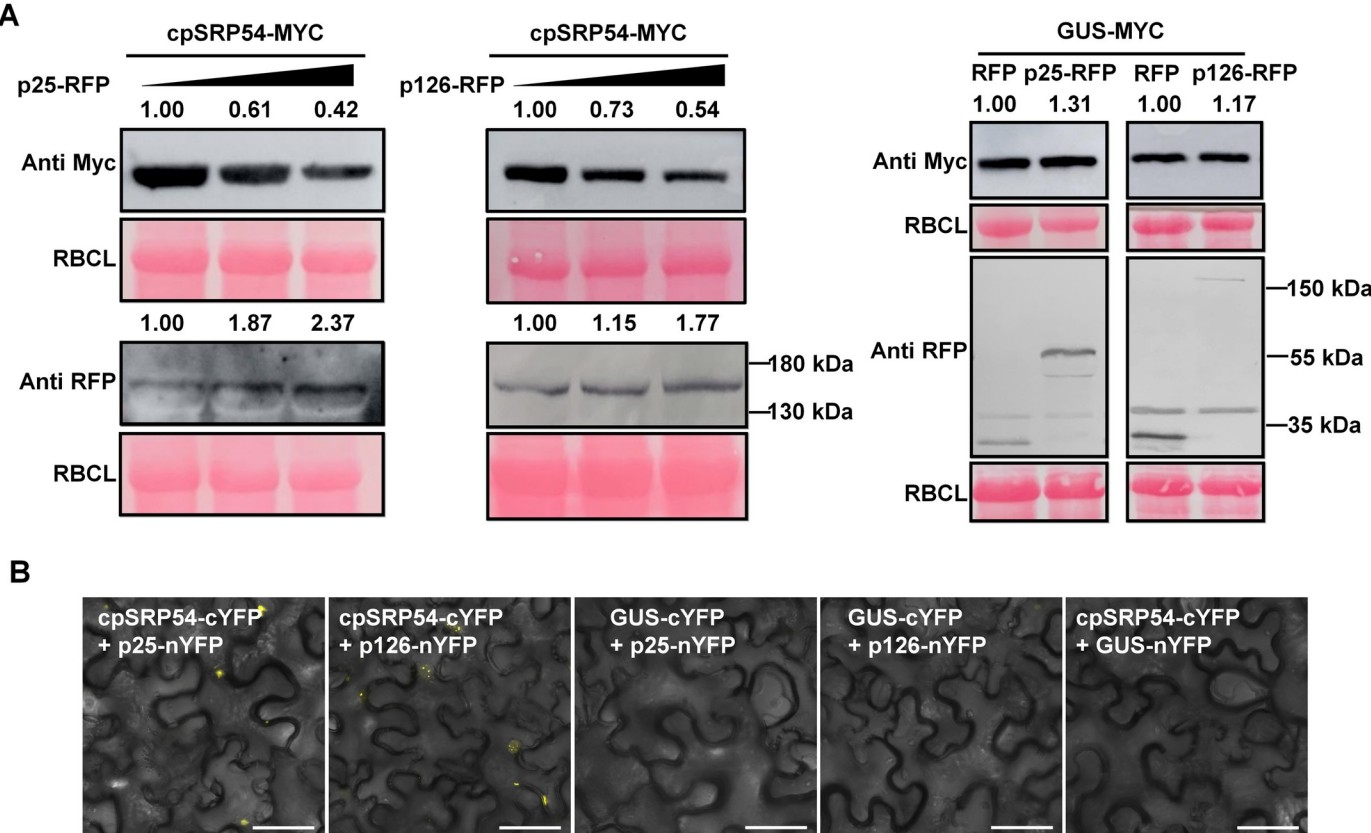

**Fig 7. PVX p25 protein and PMMoV p126 induce cpSRP54 degradation.** A. Accumulation of cpSRP54-Myc decreased with increasing amounts of p25-RFP or p126-RFP. In the presence of p25-RFP or p126-RFP, the protein levels of GUS-Myc showed no obvious change. B. BiFC assays showing the interaction of cpSRP54 with p126 and p25. Scale bar, 50 μm.

suppression of JA levels or signaling. Many viral factors have been found to interfere with JA signaling, especially the key JAZ-MYC hub, through virus-plant interactions [4]. Meanwhile, several lines of evidence indicates that the biosynthesis of JA could also be modulated by viruses: (1) RBSDV infection leads to decreased JA levels in wheat plants [12]; (2) At the late stage of RBSDV infection in rice plants, JA concentration is significantly lower than that of the uninfected control plants [10]; (3) TYLCSV C2 protein can suppress the expression of several JA biosynthesis genes [7]; (4) RRSV suppresses the expression of JA biosynthesis genes through viral-induced miRNA319 and its target gene *TCP21* [12]. However, direct evidence of how viruses regulate JA biosynthesis is still lacking. Our study here first shows that viruses trigger degradation of cpSRP54, a chloroplast protein that is responsible for delivering the JA biosynthesis enzymes AOCs to the TM where it normally functions. This directly inhibits JA biosynthesis.

For more than a decade, there has been conflicting evidence regarding the precise localization of JA biosynthesis enzymes (13LOX, AOS and AOC) in chloroplasts. In contrast to studies showing that LOX and AOS are bound to chloroplast envelopes [39–41], 13LOX and AOS from tomato, tea, and Arabidopsis were shown to be targeted to the TM [32,42]. Consistently, compelling evidence obtained from potato showed that 13LOX, AOS and AOC were all bound to the TM to varying degrees, indicating that the TM a crucial site for JA biosynthesis [31]. The results did not exclude the association of LOX and AOC to stroma, which suggested that partition of LOX and AOC between stroma and TM may be subjected to a dynamic process in

response to hitherto unknown factors. The results reported here indicate that AOC.1 in *N. benthamiana* could co-localize with a thylakoid marker and be detected in the thylakoid fraction (Fig 4A and 4D). Additionally, most AOC.1 was found localized to the TM with a smaller amount in the stroma (Fig 4B and 4C), supporting the idea that the TM is a key site for this branch of oxylipin synthesis, and that AOCs may be dynamically transported between the stroma and the TM.

In higher plants, cpSRP54 forms a signal peptide-based sorting system with cpSRP43, targeting a subset of proteins (mainly LHCPs) to the TM. After import into the chloroplast and removal of its transit peptide, LHCP binds cpSRP to form a cpSRP/LHCP transit complex. This transit complex traverses the chloroplast stroma and docks to the TM by interactions with the TM-bound cpSRP receptor cpFtsY and the integrase Alb3 [29,30,43]. In this system, cpSRP54 directly binds to cpSRP43 and its receptor cpFtsY, and LHCP is able to interact with cpSRP43 and Alb3 [29]. It is debated whether LHCP binds to cpSRP54, since their binding was observed in some studies, but not in others [44–47]. Here we demonstrate that cpSRP54 interacts with AOC.1 within chloroplasts, and is responsible for delivering AOC.1 to the TM (Figs 3 and 4). Like LHCPs, AOCs are nuclear-encoded proteins with a predicted N-terminal cleavable chloroplast transit peptide [48], but whether AOCs uses the same cpSRP54/cpSRP43/cpFtsY/Alb3 mechanism or not is not yet clear. Understanding the connection between AOCs and other elements in this process is necessary to address this issue.

cpSRP54 functions to deliver the TM-related proteins onto the membrane of TM. Suppression of cpSRP54 expression may have other effects that could possibly crosstalk with cpSRP54-mediated immunity. We therefore tried to detect other ways in which cpSRP54 might function in defense against viruses, especially through the SA or ABA pathways. Our results showed that the content of SA or ABA did not change significantly in *cpSRP54*-silenced plants, which indicates that the SA and ABA pathways are very unlikely to be cross talking with cpSRP54-mediated immunity (S3A and S3B Fig). Additionally, the expression of *13LOX* and *AOS* transcripts was clearly repressed when *cpSRP54* was silenced (S20 Fig). This suggests that cpSRP54 may also affect JA biosynthesis at the transcriptional level by regulating gene expression. Further experimental analysis is needed to illuminate any other roles of cpSRP54 in defense against viruses.

To establish efficient infection, viruses have evolved a variety of strategies to avoid or suppress host defense. Recent studies have shown that potyvirus P1 proteins play a role in countering host RNA silencing because the self-cleavage that separates it from HC-Pro activates its RNA silencing suppressor (RSS) activity [49]. In fact, P1 itself does not have RSS activity, and its real contribution in viral infection is still vague. Here we showed that P1 was able to suppress JA-mediated defense by inducing the degradation of cpSRP54 (Figs 6D–6F and S13). This post-translational regulatory activity of P1 on proteins has consistently been observed in a very recent study based on genetics and label-free proteomic approaches [50], and was further shown to be linked to the 26S proteosome and autophagy in our study, although the underlying molecular detail is still lacking (Fig 6G and 6H). P1 was localized in both nuclei and cytoplasm when expressed in plants (S21 Fig). The relationship between the cytosol-localizing 26S proteasome pathway and the chloroplast-localizing cpSRP54-P1 needs further investigation because it appears difficult to understand how the interaction might work during TuMV infection. However, several reports have indicated an association of the 26S proteasome with chloroplasts. For example, it has been reported that some E3 ligases are localized in plastids and can directly target chloroplast proteins for degradation by the 26S proteosome [51,52]. Also, some E3 ligases can ubiquitylate chloroplast proteases, thereby regulating protein degradation in chloroplasts [53]. It therefore seems possible that cpSRP54-P1 could be

associated with the 26S proteosome in such a manner. The mechanism by which cpSRP54-P1 and the 26S proteosome interact needs to be explored next.

The hypothesis that cpSRP54 acts upstream of JA biosynthesis and in the context of a relatively conserved antiviral defense role of JA, is supported by our findings that PMMoV and PVX infection could also downregulate the protein level of cpSRP54, which implies that cpSRP54 is a common target for various viruses to combat JA-mediated defense. It is notable that transcripts of *cpSRP54* were also downregulated in TuMV-infected plants, which indicates a possible mechanism by which TuMV regulates JA synthesis at the transcriptional level (S1C Fig). In addition to manipulating the JA pathway, plant viruses strike back on every aspect of plant defense, including RNA silencing, the ubiquitin proteosome system or autophagy, translation repression, other defense hormone and hormone regulatory pathways, and plant innate immunity, to win the arms race [54,55]. Some conserved core elements, such as the JAZ-MYC hub in JA signaling, rice auxin response factor 17 (OsARF17) in auxin signaling, and Argonaute 1 (AGO1) in RNA silencing defense pathway, have been shown to be convergently targeted by various viruses. Identification of more common factors will open new possibilities in the development of broad-spectrum antiviral strategies. Interestingly, the relative JA levels and the expression of *cpSRP54* and *AOCs* in plants recover at the late stages of TuMV infection (Figs 5D, S1C and S11). This may indicate a mechanism by which plants maintain the JA balance needed for survival, and which needs to be investigated next.

Most identified VSRs are multifunctional [56]. Besides being a VSR, they also function as replicases, coat proteins, proteases, movement proteins, or helper components for viral transmission, participating in the virus life cycle [57–61]. Additionally, it was shown that transgenic expression of VSRs including 2b of CMV, p25 of PVX, HC-Pro of potato virus Y (PVY; genus *Potyvirus*) and p126 of TMV inhibited the expression of JA-regulated genes [36]. We here found that p25 (from PVX) and p126 (from PMMoV) induced the degradation of cpSRP54 and inhibited JA responses, which suggests that it may not be unusual or incidental for VSRs to interfere with JA responses (Fig 7).

In summary we propose a model in which TuMV downregulates a chloroplast protein, cpSRP54, to inhibit JA biosynthesis and promote efficient viral infection (Fig 8). In this model, cpSRP54 interacts with JA biosynthesis enzyme AOCs and is responsible for delivering AOCs to the thylakoid membrane (TM) to maintain its normal function. TuMV P1 directly associates with cpSRP54 and mediates its degradation through both the 26S proteosome and autophagy pathways. Moreover, cpSRP54 interacts with AOC in Arabidopsis and rice, and is also suppressed by PMMoV and PVX, indicating that is commonly targeted by viruses to inhibit JA-mediated defense.

## Experimental procedures

### Plant materials and virus inoculation

*Nicotiana benthamiana* plants were all grown in soil at 26˚C with 16-h light/ 8-h dark cycle. Three- to four-week-old *N. benthamiana* plants were used for *Agrobacterium tumefaciens* (strain GV3101)-mediated expression as described previously [62]. For virus infection analysis, *Agrobacterium* cultures carrying different virus infectious clone (TuMV-GFP, TuMV, PVX-GFP, or PMMoV-GFP) were infiltrated into leaves. Control plants were infiltrated with empty plasmids. Virus symptoms were examined daily and GFP fluorescence of virus was observed under UV-light. All virus infection assays were repeated at least three times and each experiment included at least six plants.

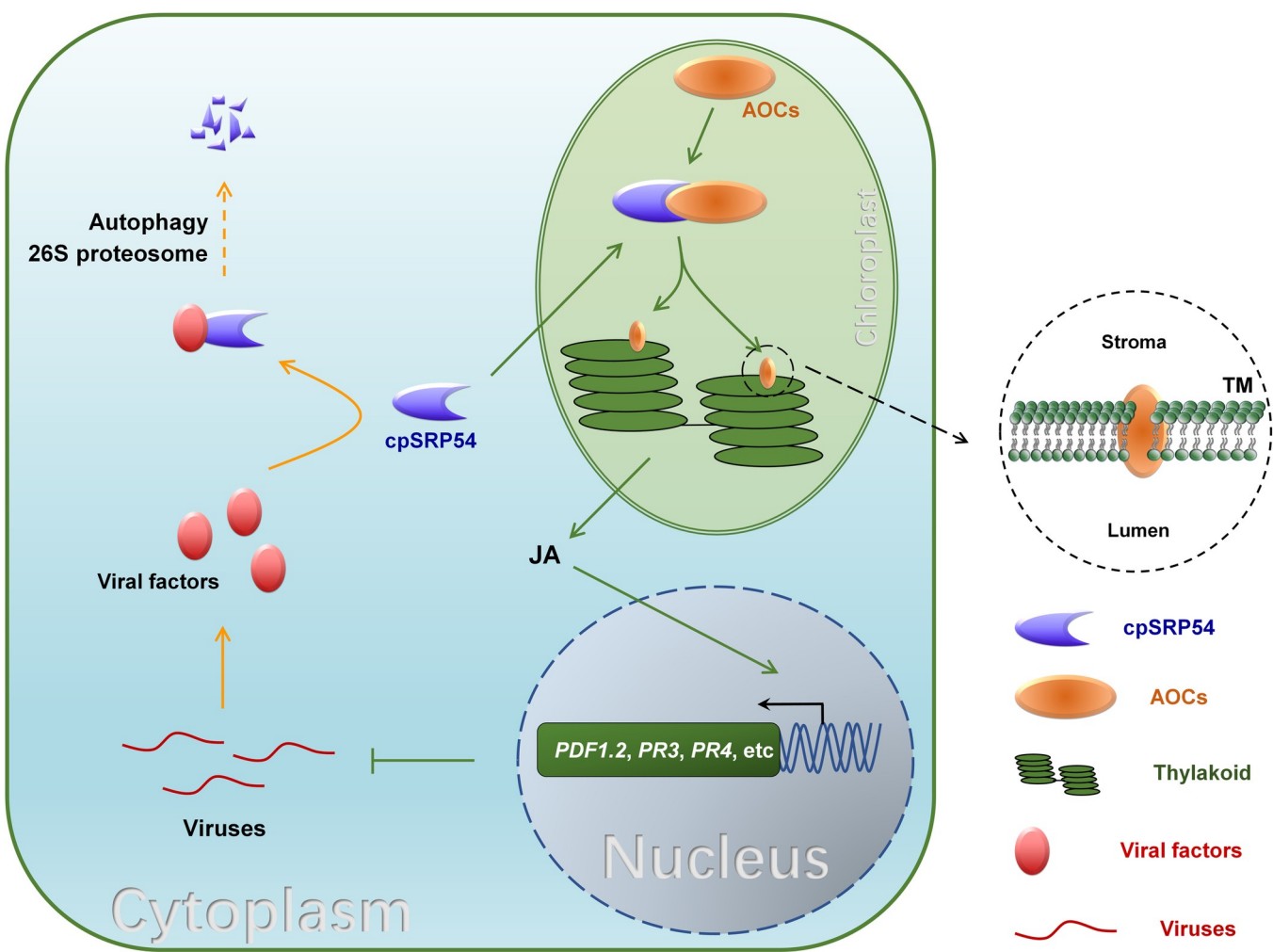

**Fig 8. A proposed working model: viral proteins suppress JA defense by degrading cpSRP54 to facilitate viral infection.** cpSRP54 is responsible for translocating AOCs to the thylakoid membrane for its function in JA biosynthesis. JA mediates plant defense against TuMV, PMMoV and PVX. Meanwhile, viral proteins, TuMV P1 protein, PMMoV p126 and PVX p25 protein interact with cpSRP54 and mediate its degradation through the 26S proteosome and autophagy pathways to suppress JA defense, hence facilitating viral infection.

## Vector construction

Gene sequences were amplified by PCR using *ExTaq* DNA Polymerase (TaKaRA). To construct transient expression vectors expressing proteins tagged at the C-terminal with GFP, mCherry, 4×Myc, 3×Flag, N-terminal half of YFP, and C-terminal half of YFP, the resulting PCR products were cloned into pJG045 (with corresponding tag), a pCMBIA 1300-based T-DNA vector [63]. Vectors pTRV1 and pTRV2-LIC were described previously [64]. For yeast two hybrid (Y2H) analysis, the corresponding PCR products were cloned into two SfiI restriction sites of the pPR3 or pDHB1 vector. For hairpin-mediated silencing, a partial sequence from NbcpSRP54 CDS (183 nt) or β-glucuronidase (160 nt) (as the control) was cloned into pFGC5941 in both sense and antisense orientations. All constructs were confirmed by DNA sequencing.

## Virus-induced gene silencing in *N. benthamiana*

The pTRV vectors used for gene silencing were kindly provided by Dr Yule Liu (Tsinghua University, Beijing, China) [65]. To silence *cpSRP54*, a 300nt-sequence was amplified with

primers NbcpSRP54-300nt-f and NbcpSRP54-300nt-r, then inserted into pTRV2-LIC expression vector (TRV: cpSRP54), and co-infiltrated with pTRV1 for VIGS. In parallel, the empty vector TRV:00 was used for the control treatment. The same method was used to silence *AOCs*. All constructs were transformed into *Agrobacterium* strain GV3101 and infiltrated into *N. benthamiana* plants as previously described [21]. The primers used for construction are listed in S1 Table.

### RNA analysis

Total RNA was extracted using TRIzol regent (Invitrogen, now ThermoFisher Scientific, https://www.thermofisher.com) according to the manufacturer's protocol. For quantitative RT-PCR, a LightCycler 480 Real-Time PCR System (Roche, https://www.roche.com) was used for the reaction and the results were analyzed by the $\Delta\Delta C_T$ method. *N. benthamiana actin* gene was used as the internal reference control for analysis [66]. Biological triplicates with technical replicates were performed. The primers used for RT-PCR of silencing pathway-related genes and JA-related genes are listed in S1 Table.

### Protein analysis

Total proteins for western blot (WB) assay were extracted from leaf tissues or isolated protoplasts or thylakoids as described [19,67]. Proteins were detected by specific antibodies against cpSRP54 (Hangzhou Huaan Biotechnology Co., Ltd., (HuaBio), https://www.huabio.com), TuMV CP (produced by the author's lab), PVX CP (produced by the author's lab), PMMoV CP (produced by the author's lab), plastocyanin (PC) (PhytoAB, http://www.phytoab.com), UDP-glucose pyrophosphorylase (UGPase) (PhytoAB, http://www.phytoab.com), β-Actin (ABclonal), RFP (ABclonal), GFP (TransGen, HT801) or Myc (TransGen, HT101-01) and were visualized using nitrotetrazolium blue chloride/5-bromo-4-chloro-3-indolyl phosphate (NBT/BCIP) buffer (Sigma-Aldrich, https://www.signmaaldrich.com) or enhanced chemiluminescence reaction, ECL (Transgene Biotech, Beijing, China, https://www.transgenbiotech.com).

For Co-immunoprecipitation (Co-IP) assays, total proteins were extracted at 60–72 hpi with ice-cold GTEN extraction buffer (25 mM Tris -HCl, pH7.5, 1 mM EDTA, 10% glycerol, and 150 mM NaCl), 10 mM DTT, 1 mM PMSF, 0.15% Nonidet P40 and 1 × protease inhibitor cocktail (Roche) [68], and incubated with GFP-Trap_MA beads (Chromotek) for 1 h at room temperature. The beads were collected by brief centrifugation and washed at least four times in lysis buffer, and then immunoblotted with GFP, Flag or Myc antibodies.

### Detection of hormone content

*N. benthamiana* leaf tissues (~800 mg) were analyzed by high-performance liquid chromatography tandem mass spectrometry (HPLC-MS/MS) with JA-type, SA type, or ABA type samples (Sigma-Aldrich) according to a method previously described [69]. Three independent replicates each containing three biological repeats were used for hormone quantification. Hormone levels were measured by Zoonbio Biotechnology Co., Ltd and RUIYUAN Biotechnology Co., Ltd.

### Plant hormone treatment

The *N. benthamiana* leaves were sprayed every 2 d with 50 μM MeJA for 7 d or 8 d with 0.1% ethanol as a control. 24 h after the first treatment of MeJA, plants were inoculated with TuMV-GFP.

## Yeast two-hybrid assay

All work with yeast was done using the yeast strain NMY51. The yeast constructs were co-introduced into NMY51 by LiOAc-mediated transformation as previously described [70]. Yeast cells were plated on SD (-Trp/-Leu) to test if a good transformation efficiency had been achieved, and were then plated on selected SD (-Ade/-His/-Leu/-Trp) to analyse interactions between the expressed proteins.

## Laser confocal microscopy and transmission electron microscopy assay

The *N. benthamiana* leaf tissues or protoplasts expressing proteins were imaged using a Lecia TCS SP5 confocal microscope at 48–72 hpi. The full-length coding regions of proteins were fused either to the N-terminal (nYFP) or C-terminal (cYFP) half of YFP, or green fluorescent protein (GFP), or mCherry for the bimolecular fluorescence complementation (BiFC) or co-localization assays.

For immunocytochemistry, small leaf tissues (1 mm × 3 mm) were sampled, fixed and embedded as described previously [31]. Thin sections were labelled with anti-GFP antibody followed by gold-conjugated anti-mouse antibody, and analyzed on a transmission electron microscopy.

## Isolation of protoplasts and thylakoids

Protoplast isolation was done as previously described [19]. The purified intact protoplasts were diluted appropriately and counted with a hemocytometer under a microscope for proto-plast yield. Intact chloroplasts were isolated and purified on Percoll gradients [71]. Thylakoids from intact chloroplasts were isolated using the methods described by [72]. Chloroplasts were ruptured by osmotic shock in 50 mM Tris-HCl, pH 8.0, and 5 mM $MgCl_2$ for 10 min. Then the thylakoid fraction was collected, washed and resuspended in thylakoid resuspension buffer (10 mM Tris-HCl, pH 8.0, and 5 mM $MgCl_2$). Sequential sonication and centrifugation were used to liberate the thylakoid membrane proteins. All the procedures were done at 4˚C and all solutions contained 1 × protease inhibitor cocktail (Roche).

## MG132 and 3-MA treatment

Phosphate-buffered saline containing 2% dimethyl sulfoxide (DMSO, as control) or an equal volume of DMSO with 100 μM MG132 (Sigma), or $H_2O$ as a control and an equal volume of $H_2O$ with 10 mM 3-MA (Sigma), was infiltrated into the leaves that pre-agroinfiltrated with targeted proteins [35]. After 16 h, samples were collected.

## Supporting information

**S1 Fig. RNA expression level of *cpSRP54* in TuMV-infected *N. benthamiana*.** A. Amino acid sequence alignment of cpSRP54s from *N. benthamiana*, rice and Arabidopsis. B. Amino acid (numbers shadowed with light pink) and nucleotide identities (numbers shadowed with light blue) between the cpSRP54s. C. Quantification of *cpSRP54* mRNA levels in TuMV-infected plants within 14 dpi by qRT-PCR analysis. Means ± SD values are from three independent plants per treatment.
(TIF)

**S2 Fig. Silencing of *cpSRP54* by RNAi construct in *N. benthamiana*.** A. Phenotype of leaves inoculated with *GUS* RNAi construct (as control) and *cpSRP54* hairpin RNAi construct at 3 dpi. Silencing of *cpSRP54* did not cause obvious chlorosis. B. *cpSRP54* mRNA level analysis by

qRT-PCR in *cpSRP54*-silenced plants compared to control plants at 3 dpi. Means ± SD values are from three independent plants per treatment and were normalized against *NbActin*. **, P<0.01 according to Student's *t*-test. C. TuMV-GFP infection in plants pretreated with *GUS* RNAi construct and *cpSRP54* RNAi construct. Plants were photographed under UV at 7 dpi. Viral CP accumulation in systemic leaves was determined by WB. Actin served as a loading control. This experiment was repeated at least three times, and one representative result is shown. The protein levels were quantified by ImageJ. D. Relative viral RNA levels quantified by qRT-PCR. Means ± SD values are from three independent plants per treatment. **, P<0.01 according to Student's *t*-test.
(TIF)

**S3 Fig. Relative levels of endogenous SA (A) and ABA (B) in TRV:00 and TRV:cpSRP54 treated plants by LC-MS at 12 dpi.**
(TIF)

**S4 Fig. MeJA treatment alleviated the susceptibility of *AOCs*-silenced plants to TuMV.** A. Phenotype in TRV:00 and TRV:AOCs treated plants at 12 dpi. B. Effect of MeJA treatment on TuMV-infection in plants inoculated with TRV:00 or TRV:AOCs at 7 dpi. Plants were photographed under UV light. Bars, 2 cm. C. Accumulation of viral CP protein quantified by WB. Ponceau S-stained RBCL was used as a loading control. Tests were performed independently three times with similar results. D. Quantification of viral RNA levels by qRT-PCR. Means ± SD values are from three independent plants per treatment. **, P<0.01, *, P<0.05 according to Student's *t*-test.
(TIF)

**S5 Fig. The interaction analysis of cpSRP54 and AOC.2 by Co-IP (A) and BiFC (B) assays.** Scale bar, 10 μm.
(TIF)

**S6 Fig. Accumulation of AOC.1-GFP was not affected by co-expression of cpSRP54-Myc.** GUS-Myc and empty GFP were used as controls. Ponceau S-stained RBCL was used as a loading control. The protein levels were quantified by ImageJ. Tests were performed independently three times.
(TIF)

**S7 Fig. Sequence analysis of AOCs from different plants.** A. Amino acid sequence alignment of AOCs from *N. benthamiana*, rice and Arabidopsis. B. Amino acid (numbers shadowed with light pink) and nucleotide identities (numbers shadowed with light blue) among NbAOC.1, NbAOC.2, OsAOC, AtAOC1, AtAOC2, AtAOC3 and AtAOC4.
(TIF)

**S8 Fig. Interaction between cpSRP54 and AOC in Arabidopsis (A) and rice (B) by BiFC assay.** cpSRP54s and AOCs were respectively fused to the C-terminal (cYFP) and N-terminal (nYFP) half of YFP. Confocal imaging was performed at 2 dpi. Scale bar, 10 μm.
(TIF)

**S9 Fig. BiFC analysis with cpSRP54-cYFP and AOC.1-nYFP in TuMV-infected plants at 2 dpi.** Scale bar, 10 μm.
(TIF)

**S10 Fig. WB analysis confirming the transient expression of cpSRP54-Myc.**
(TIF)

**S11 Fig. qRT-PCR analysis of *AOCs* transcripts in TuMV-infected plants within 14 dpi.** Means ± SD values are from three independent plants per treatment. (TIF)

**S12 Fig. Results of qRT-PCR analysis showing that *cpSRP54* transcripts were not affected by expression of P1-GFP.** Relative expression level of *cpSRP54* in *N. benthamiana* plants transiently expressed GFP or P1-GFP at 2 dpi. Bars represent the standard errors of the means from three biological repeats. *NbActin* was used as the internal control. (TIF)

**S13 Fig. Confocal microscopy analysis showing the localization of AOC.1-GFP within chloroplasts in the presence of P1-Myc at 3 dpi.** Bars, 10 μm. (TIF)

**S14 Fig. Confocal microscopy analysis showing the localization of AOC.2-GFP within chloroplasts in the context of P1-Myc expression (A) and TuMV infection (B) at 3 dpi.** (TIF)

**S15 Fig. Control experiments in P1-mediated cpSRP54 degradation.** A and B. Accumulation of cpSRP54-Myc was not affected by empty GFP (A), or CP-GFP (B) whether treated with MG132 and 3-MA or not. B. P1-GFP did not affect accumulation of GUS-Myc, whether treated with MG132 and 3-MA or not. (TIF)

**S16 Fig. qRT-PCR analysis to confirm the silencing efficiency of *ATG5*, *ATG7*, *RPN10*, and *RPN13*.** The leaf samples were harvested at 14 dpi. Values represent the Means ± SD. **, $P<0.01$ according to Student's *t*-test. (TIF)

**S17 Fig. The thylakoid distribution of AOC.1-GFP (A) and JA level (B) were restored when P1 was co-expressed in *RPN10*-silenced plants.** (A) Confocal microscopy analysis showing the localization of AOC.1-GFP transiently expressed in TRV:00 or TRV:RPN10 treated plants when co-expressed with P1-Myc at 3 dpi. (B) The relative JA level in the above plants. Three independent replicates each containing three biological repeats were used for hormone quantification. *, $P<0.05$ according to Student's *t*-test. (TIF)

**S18 Fig. qRT-PCR analysis showing the mRNA levels of *cpSRP54*, *PR3* and *PR4* in the presence of p25-RFP (A) or p126-RFP (B).** (TIF)

**S19 Fig. Downregulation of cpSRP54 in PMMoV or PVX-infected plants impairs JA-mediated defense against viral infection.** A. The accumulation of cpSRP54 proteins in PMMoV or PVX-infected *N. benthamiana* was determined by WB. B. PMMoV-GFP or PVX-GFP were inoculated onto *cpSRP54*-silenced (TRV:cpSRP54), non-silenced, CK (0.1% ethanol) treated wild type and MeJA (50 μM) treated wild type plants. GFP fluorescence on the newly-emerged leaves indicate systemic infection by viruses at 7 dpi. Plants were photographed under UV light. C. PMMoV and PVX CP antibody were used to detect the accumulation of PMMoV-GFP and PVX-GFP, respectively, in plants at 7 dpi. The protein levels were quantified by ImageJ and normalized against actin protein levels. These experiments are the representatives of three independent biological experiments with similar results. (TIF)

**S20 Fig. qRT-PCR analysis showing the mRNA levels of *13LOX* and *AOS* in *cpSRP54* silenced *N. benthamiana*.**
(TIF)

**S21 Fig. Confocal analysis of transiently expressed P1-GFP in *N. benthamiana*. Bars, 50 μm.**
(TIF)

**S1 Table. Primers used for analysis.**
(DOCX)

**S2 Table. Statistics to support Fig 4C.**
(DOCX)

# Acknowledgments

We thank Prof. M. J. Adams, Minehead, UK, for correcting the English of the manuscript. We thank Dr. Fernando Ponz for providing the TuMV infectious clone, Dr. Stuart MacFarlane for providing the PVX infectious clone.

# Author Contributions

**Conceptualization:** Fei Yan.

**Formal analysis:** Fei Yan.

**Funding acquisition:** Jianping Chen, Fei Yan.

**Investigation:** Mengfei Ji, Kelei Han, Weijun Cui, Xinyang Wu, Binghua Chen, Yuwen Lu, Jiejun Peng, Hongying Zheng, Shaofei Rao, Guanwei Wu, Fei Yan.

**Methodology:** Mengfei Ji, Jinping Zhao, Kelei Han, Weijun Cui, Xinyang Wu, Binghua Chen, Yuwen Lu, Jiejun Peng, Hongying Zheng, Shaofei Rao, Guanwei Wu.

**Project administration:** Jianping Chen, Fei Yan.

**Supervision:** Jianping Chen, Fei Yan.

**Writing – original draft:** Mengfei Ji, Fei Yan.

**Writing – review & editing:** Jianping Chen, Fei Yan.

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
