## [Decision Letter · Decision Letter 0]

5 Jun 2021

Dear Dr. Yan,

Thank you very much for submitting your manuscript "Viral Proteins Suppress JA Biosynthesis by Degrading cpSRP54 that Delivers AOC onto the Thylakoid Membrane to Facilitate Viral Infection" for consideration at PLOS Pathogens. As with all papers reviewed by the journal, your manuscript was reviewed by members of the editorial board and by several independent reviewers. In light of the reviews (below this email), we would like to invite the resubmission of a significantly-revised version that takes into account the reviewers' comments.

All reviewers raise concerns regarding the observed interaction between AOC with spSRP54. The reviewers would like to see this interaction in the context of the viral infection. In addition, specificity of cpSRP54 knockdown on TuMV infection should be addressed. Reviewers also raised concerns regarding image quality and lack of appropriate controls in many of the experiments. We therefore ask you to address these issues with additional work.

We cannot make any decision about publication until we have seen the revised manuscript and your response to the reviewers' comments. Your revised manuscript is also likely to be sent to reviewers for further evaluation.

Sincerely,

Savithramma P. Dinesh-Kumar

Associate Editor

PLOS Pathogens

Shou-Wei Ding

Section Editor

PLOS Pathogens

Kasturi Haldar

Editor-in-Chief

PLOS Pathogens

orcid.org/0000-0001-5065-158X

Michael Malim

Editor-in-Chief

PLOS Pathogens

orcid.org/0000-0002-7699-2064

Reviewer's Responses to Questions

**Part I - Summary**

Reviewer #1: Recent studies have demonstrated that genetic and chemical manipulation of JA can significantly alter plant-virus interactions. The authors previous work demonstrated the cpSRP54 protein was significantly reduced by RSV infection in N. benthamiana. In this manuscript the authors attempt to dissect the role of JA and cpSRP54 in plant-TuMV interactions. This is an exciting and important area of research where many questions remain. The authors demonstrate here:

1. cpSRP54 protein levels are reduced in N. benthamiana during TuMV infection and silencing cpSRP54 increased TuMV CP levels (Fig S2; Fig 1A)

2. Silencing cpSRP54 decreased JA and JA-related transcript abundance (Fig 1A-C, Fig. S3C).

3. Spraying with MeJA decreased TuMV CP protein levels (Fig. S5)., while silencing ACO (related to JA biosynthesis) increased TuMV CP abundance (Fig 6D)

4. cpSRP5 interacted with AOC (Y2H, Co-IP, and BIFC) in N. benthamiana (Fig. 2) and rice and Arabidopsis (BIFC; Fig S9)

5. Full AOC localization in the thylakoid membrane requires cpSRP54 (Fig 3).

6. During TuMV infection AOC localization in protoplast is changed (Fig. 4A).

7. P1 and cpSRP54 interact (BIFC, co-IP, Y2H) (Fig. 5A,B,C)

8. P1 decreases cpSRPR54 protein levels (Fig 5D) and JA levels (Fig S12 B)

9. Other viruses (PMMoV and PVX) and their silencing suppressors reduce cpSRP54 (Fig. 6A; Fig. S16 A).

10. Silencing cpSRP54 increases CP levels for PMMoV and PVX and treating with MeJA decreases CP for these viruses.

The authors did a great job with most of the experiments and the paper advances our understanding of the importance of cpSRP54 in TuMV-plant interactions. However, I have a few concerns that need to be addressed before this manuscript is ready for publication. See comments below.

Reviewer #2: Ji et al. discovered that TuMV P1 protein interacted with the chloroplast protein cpSRP54 and mediated its degradation through the 26S proteosome and autophagy pathways during the viral infection. This would lead to block the delivery of the allene oxide cyclases (AOCs), key Jasmonic acid biosynthesis enzymes, onto the thylakoid membrane (TM). Since JA is a crucial hormone in plant antiviral immunity, the failure of JA biosynthesis could impair plant immunity and enhance viral infection. The authors suggested that this may be a common mechanism used by viruses to counter the antiviral JA pathway. It is an interesting work, however, a few aspects should be addressed.

1. cpSRP54 responds to deliver all the thylakoid membrane proteins, including AOCs and all the photosynthesis-related proteins. Thus, its knockdown could have a great effect on the efficiency of photosynthesis. The knockdown plants showed a chlorosis phenotype as expected (shown in Fig. S2). The relative expression of cpSRP54 is about 5% left in the knockdown plants. However, using the hairpin construct to knock down the gene expression has also shown about 6% left but not chlorosis. How can this explain the relationship of this protein cpSRP54 responding to photosynthesis with the phenotype?

2. The relative accumulation level of CP on the western blots, as an example shown in Figure 1, that using RBCL as the loading control is not a fair comparison. The phenotype of the knockdown plant is in a chlorosis phenotype that the expression level of RBCL is no longer a good marker used as a loading control.

3. Because cpSRP54 responds to deliver the TM-related proteins on the membrane of TM, the effects on the reduced expression could be in all aspects. How can the authors exclude the effect derived from other proteins beyond the AOCs? Although the authors did exclude the effect derived from chlorosis and exogenous application of MeJA alleviated the susceptibility of AOC-knockdown plants to TuMV. Did the AOCs-knockdown plants shown any particular phenotype?

4. Line 199, “However, in cpSRP54-silenced cells, the fluorescence from AOC.1-GFP was much more diffused (Fig 3A).” The word “diffused” in the description is vague, hard to make a comparison.

5. The resolution of the immunolocalization under electron microscopy was not clear. It is hard to see the gold particles on the figure. Could it be replaced with a better one?

6. In Figure 3D, the authors indicated that the protein AOC.1-GFP in the thylakoid fraction of cpSRP54-silenced cells was much less than that in non-silenced cells. This description required control to reveal that the level of thylakoid marker was not changed in the knockdown plants. The authors used the anti-PC as the control marker. However, the signal of PC banding in the western blot is not clear. Whether using AtOE23-mCherry used in Figure 3A is better in this experiment?

7. The data in Figure 4C was not matched to the description in line 228 that the relative level in the infected plants declined continuously, indicating that TuMV suppressed JA biosynthesis (Fig 4C). By contrast, the level of JA in the infected plants is higher than that in the mock-inoculated plants shown in figure 4C. There must be some mistake or wrong figure used in this result.

8. The interaction of cpSRP54 and P1 was revealed by Y2H, co-IP, and BiFC. However, the confocal result of BiFC was not clear might due to the resolution was not high enough. The results indicated that the interaction of P1-nYFP and spSRP54-cYFP was co-localized with chloroplast. Is this true shown in Figure 5A? However, the 26S proteasome is in the cytosol, how can this interaction work during TuMV infection?

9. The results of treating the leaves with MG132 or 3MA that restored the level of cpSRP54 shown in Figure 5E and concluded cpSRP54 could be degraded by proteasome and autophagy pathways were vague. The virus infection or agroinfiltration could induce autophagy that targets the chloroplast (chlorophagy), the contents inside the chloroplasts (cpSRP54 might be included) were engulfed to chlorophagy for degradation in the vacuole. Therefore, blocking the autophagy pathway could also prevent the degradation of cpSRP54 in the chloroplast possibly. This hypothesis is not necessary to go through the interaction with P1.

Reviewer #3: In this manuscript, Ji et al. report an interesting story of viral proteins suppress JA biosynthesis to facilitate viral infection. The authors found that viral proteins suppress JA-mediated defense by degrading cpSRP54 to inhibit the delivering of AOCs onto the thylakoid, which resulting in the decrease of JA biosynthesis. Overall, this manuscript expands on our current knowledge regarding how virus overcome the host antiviral immune response.

**Part II – Major Issues: Key Experiments Required for Acceptance**

Reviewer #1: 1. AOC interactions with cpSRP54 and localization in the thylakoid membrane needs to be shown in the viral context. AOC immunolocalization in TuMV infected plants needs to be conducted or immunoblots need to be conducted with thylakoids from mock and TuMV infected plants, probing against AOC.

2. AOC localization in the thylakoid membrane needs to be shown in the P1 protein context. The authors show AOC synthase accumulation is reduced in thylakoid membranes in the plants that were infiltrated with antisense cpSRP54 using immunolocalization and immunoblots, but it is very important to show the same thing happening in plants that are infected with TuMV (see comment 1) and in plants that are transiently expressing P1 only to fully support their major claims.

3. No quantitative measures of viral titer are conducted to demonstrate significant differences. Instead CP protein levels were measured, which are not quantitative, and differences in CP levels are very minor in several important figures (Fig 1, Fig S6E, Fig. S3C. Fig 6D). For example, qRT-PCR needs to be conducted for TuMV CP transcripts for the experiments in Fig 1 D to clearly demonstrate significant changes in TuMV titer or lack of. Differences in CP levels are hard to differentiate and this experiment is very important. Adding qRT-PCR data in other experiments were differences in CP protein levels were difficult to resolve would significantly strengthen the conclusions (Fig 1, Fig S6E, Fig. S3C. Fig 6D).

Reviewer #2: (No Response)

Reviewer #3: However, before acceptance for publishing in PloS Pathogens, the following concerns need to be addressed.

1 The author concluded that the decrease of JA biosynthesis due to the abnormal delivering of AOCs onto the thylakoid. They should detect the expression level of AOCs and the protein level of AOCs post infected with virus. Whether the interaction of cpSRP54 and AOCs influence the stability of AOCs.

2 They believed that “When AOC.1-GFP was co-expressed with cpSRP54-Myc in TuMV225 infected plants, the normal distribution of AOC.1 was partially restored.”. However, I can not see any difference between cpSRP54-Myc and mock. They should provide more evidences.

3 The protein level of AOCs in the thylakoid was just detected in TRV-cpSRP54. To support the conclusion that virus inhibits the delivering of AOCs onto the thylakoid, the protein level of AOCs in the thylakoid should be also measured in virus infected and P1-Myc plants.

**Part III – Minor Issues: Editorial and Data Presentation Modifications**

Reviewer #1: 4. The claim that TuMV alters JA content is important, but details on methods are missing making Fig 4C making it difficult to interpret. How were these plants inoculated and how were “Mock” plants treated? JA is damage induced and increases in JA in early time points and decreases in late time points could be due inoculation damage if the mocks were not also damaged in the same way.

5. The title should be revised to focus on P1, as the authors have only shown convincingly that P1 degrades plant protein cpSRP54 that prevents delivery of AOC to thylakoid and suppress JA biosynthesis. Although the authors show that two viral proteins p25 and p126 interacts with cpSRP54, the subsequent steps of that interaction suppressing JA biosynthesis does need to be demonstrated or the title revised.

6. The other alternate hypothesis that is not properly addressed in the manuscript is the possibility of some other protein or mechanism that might crosstalk with cpSRP54’s removal by P1 to alter viral titer. A quick search shows cpSRP54 Arabidopsis mutants have altered photosynthesis, ROS production, ABA, and carotenoids. Is it possible one of these changes may be mediating impacts on plant- virus interaction and may be more important for cpSRP54-TuMV interactions?

7. Is it possible that removal of cpSRP54 may hamper JA biosynthesis at a step before AOC synthase or after it? I think that measuring JA, AOC thylakoid localization, or possibly the substrate for AOC synthase in plants that were infiltrated with inhibitors of proteasome and autophagy or in the RPN11/ATG5, AT7 silenced plants, expressing cpSRP54 and PI may be a definitive step towards showing that P1 degrades cpSRP54, which stops AOC synthase movement and that is the exact step which is targeted by the viral effector to reduce JA biosynthesis.

8. Some of the figures in supplemental are quite important and should be moved to the main text of the manuscript. For example, Fig S2, Fig. S5, Fig S6a-D and S12 B should be in the main text. The results section in lines 149 to 180 solely relies on supplemental figures which shows how important some of these figures are that are in supplemental and should be moved to the main text.

9. The methods section is a bit too terse. Please mention important details such as how many biological replicates were used for each expt. This information is sporadically given for the experiments performed. How many plants were used for real time PCRs? What were their age? How many biological replicates were used for the immunoblot analyses. How many times were the BiFC experiments repeated/done? The images shown are from one plant, but did the authors confirm the same result in different biological replicates? If so, how many? Please give more details about what plasmids were used for the BiFC constructs.

Reviewer #2: 1. According to ICTV definition, a species name is written in italics with the first word beginning with a capital letter.

Reviewer #3: (No Response)

PLOS authors have the option to publish the peer review history of their article (what does this mean?). If published, this will include your full peer review and any attached files.

Reviewer #1: No

Reviewer #2: No

Reviewer #3: No
---

## [Decision Letter · Decision Letter 1]

21 Sep 2021

Dear Dr. Yan,

Thank you very much for submitting your manuscript "Turnip mosaic virus P1 suppresses JA biosynthesis by degrading cpSRP54 that delivers AOCs onto the thylakoid membrane to facilitate viral infection" for consideration at PLOS Pathogens. As with all papers reviewed by the journal, your manuscript was reviewed by members of the editorial board and by several independent reviewers. The reviewers appreciated the attention to an important topic. Based on the reviews, we are likely to accept this manuscript for publication, providing that you address the reviewer's comments.

Specifically, the revised manuscript should include a time course data on cpSRP54 transcript level and JA content after TuMV infection. Other comments could be addressed by editing the text.

Please prepare and submit your revised manuscript within 60 days. If you anticipate any delay, please let us know the expected resubmission date by replying to this email.

Sincerely,

Savithramma P. Dinesh-Kumar

Associate Editor

PLOS Pathogens

Shou-Wei Ding

Section Editor

PLOS Pathogens

Kasturi Haldar

Editor-in-Chief

PLOS Pathogens

orcid.org/0000-0001-5065-158X

Michael Malim

Editor-in-Chief

PLOS Pathogens

orcid.org/0000-0002-7699-2064

Reviewer Comments (if any, and for reference):

Reviewer's Responses to Questions

**Part I - Summary**

Reviewer #3: This is an improved version and has addressed all my concerns. I do not have further comments.

Reviewer #4: The manuscript entailed “Turnip mosaic virus P1 suppresses JA” by Ji et al reported their discovery that TuMV infection inhibits the accmulation of cpSRP54，a chloroplast signal recognition particle subunit which is required for photosynthetic proteins to target in to thylakoid membranes. Further they showed that cpSRP54 interacts with allene oxide cyclases(AOCs) which participates in JA biosynthesis. Interestingly, they demonstrated that TuMV P1 interacts with cpSRP54 and mediated the degradation of cpSRP54 via 26S proteasome and results in the reduction of JA synthesis, further benefits viral infection. More interestingly, they showed evidences that this strategy may be applied by different RNA viruses such as PVX and TMV. The story is very interesting and the figures are well illustrated. The two reviewers have raised a number of reasonable questions, and the authors have substantially addressed their concerns. However, after carefully read the manuscript, I still have some questions and there are some minor errors should be corrected before acceptance for publication. I have listed below.

Reviewer #5: In this article, Ji et al. have described an interesting viral counter-defence strategy to overcome antiviral JA response. This study is interesting and will add on to our current understanding of plant defence and viral counter-defence. However, there are some major and minor concerns which need to be addressed before the manuscript is accepted for publication.

**Part II – Major Issues: Key Experiments Required for Acceptance**

Reviewer #3: (No Response)

Reviewer #4: Major: I would happy to see if the authors could add a time course of results of JA content corelated with cpSRP54 and AOCs expressions after TuMV infection.

Reviewer #5: (1) Authors need to check JA level beyond 9.5 days post inoculation (dpi) also (at least up to 21 dpi).

(2) Author need to provide information on check the cpSRP54 transcript abundance during TuMV pathogenesis at different dpi.

(3) Expression of other JA biosynthetic genes which function upstream of AOCs (specifically AOS and LOX genes) should be checked in cpSRP54 silenced plants.

(4) What is the subcellular localization of TuMV encoded P1 protein?

(5) Authors must put effort to identify the specific domain or amino acid stretch of P1 protein responsible for the interaction with cpSRP54. This reader is of the opinion that authors must express a P1 mutant unable to interact with cpSRP54 and validate their observation by comparing with the wild-type viral protein.

(6) It will be interesting to know the impact of the interaction between cpSRP54 and AOC.2 in TuMV pathogenesis. Whether it follows the trend of AOC.1 or not needs to be investigated.

**Part III – Minor Issues: Editorial and Data Presentation Modifications**

Reviewer #3: (No Response)

Reviewer #4: Minor:

1. Some information for Ref. 21 and 22 is missing.

2. There is a most recent review paper about the interactions between host hormones and plant viral infection should be cited. See PLoS Pathogens 2021,17(2):e1009242.

3. Some figure legends are not matched with figures, such as Fig. 1C, 1D, 1F and 1G. Please check them carefully.

4. The model, some steps such as JA activates which defense genes? How cpSRP54 exits from chloroplast and be degraded is not fully understood. Therefore, the lines should use dashed lines.

5. Some paragraph on the introduction of cpSRP54 should be more to introduction.

Reviewer #5: (1) Line 25: biosynthesis should also be mentioned here along with signaling.

(2) Line 28-29: Intended meaning not clear !

(3) Line 73, 82: rice infecting viruses NOT of rice viruses.

(4) Line 90: ‘and’ would be more suitable word than ‘or’ here.

PLOS authors have the option to publish the peer review history of their article (what does this mean?). If published, this will include your full peer review and any attached files.

Reviewer #3: No

Reviewer #4: No

Reviewer #5: No

Figure Files:

Data Requirements:

Reproducibility:

References:

---

## [Editor Report · Decision Letter 2]

11 Nov 2021

Dear Dr. Yan,

We are pleased to inform you that your manuscript 'Turnip mosaic virus P1 suppresses JA biosynthesis by degrading cpSRP54 that delivers AOCs onto the thylakoid membrane to facilitate viral infection' has been provisionally accepted for publication in PLOS Pathogens.

Best regards,

Savithramma P. Dinesh-Kumar

Associate Editor

PLOS Pathogens

Shou-Wei Ding

Section Editor

PLOS Pathogens

Kasturi Haldar

Editor-in-Chief

PLOS Pathogens

orcid.org/0000-0001-5065-158X

Michael Malim

Editor-in-Chief

PLOS Pathogens

orcid.org/0000-0002-7699-2064
---

## [Editor Report · Acceptance letter]

26 Nov 2021

Dear Dr. Yan,

We are delighted to inform you that your manuscript, "Turnip mosaic virus P1 suppresses JA biosynthesis by degrading cpSRP54 that delivers AOCs onto the thylakoid membrane to facilitate viral infection," has been formally accepted for publication in PLOS Pathogens.

Best regards,

Kasturi Haldar

Editor-in-Chief

PLOS Pathogens

orcid.org/0000-0001-5065-158X

Michael Malim

Editor-in-Chief

PLOS Pathogens

orcid.org/0000-0002-7699-2064